# Impact of 3D groundwater dynamics on heat events in historical regional climate simulations over Europe

Liubov Poshyvailo-Strube[1,2], Niklas Wagner[1,2], Klaus Goergen[1,2], Carina Furusho-Percot[3], Carl Hartick[1,2,4], and Stefan Kollet[1,2]

[1]Institute of Bio- and Geosciences: Agrosphere (IBG-3), Forschungszentrum Jülich GmbH, Jülich, Germany
[2]Centre for High-Performance Scientific Computing in Terrestrial Systems (HPSC TerrSys), Geoverbund ABC/J, Jülich, Germany
[3]National Research Institute for Agriculture, Food and Environment (INRAE), Avignon, France
[4]Jülich Supercomputing Centre (JSC), Forschungszentrum Jülich GmbH, Jülich, Germany

**Correspondence:** Liubov Poshyvailo-Strube (l.poshyvailo@fz-juelich.de)

**Abstract.** The representation of groundwater processes is simplified in most regional climate models (RCMs), potentially leading to biases in simulated heat waves. Here, we introduce a unique dataset from the regional Terrestrial Systems Modelling Platform (TSMP) forced by Max Planck Institute Earth System Model at Low Resolution (MPI-ESM-LR) boundary conditions for a historical time span in the context of dynamical downscaling of global climate models (GCMs) for climate change studies.
TSMP explicitly represents 3D subsurface and groundwater hydrodynamics together with overland flow, closing the water and energy cycle from the bedrock to the top of the atmosphere. We perform an analysis of summer heat events (i.e. a series of consecutive days with a near-surface temperature exceeding the 90th percentile) for the historical time period 1976-2005 relative to the reference period 1961-1990 in a TSMP climate change scenario control run. For comparison, the analysis is repeated for an ensemble of GCM-RCM simulations with simplified groundwater dynamics from the Coordinated Regional
Climate Downscaling Experiment initiative for the European domain (EURO-CORDEX).

While our results show that TSMP simulates heat events consistently with the CORDEX ensemble, there are some systematic differences that we attribute to the representation of groundwater in TSMP. Compared to the CORDEX ensemble, TSMP simulates lower means and lower interannual variability in the number of hot days (i.e., days with a near-surface temperature exceeding the 90th percentile) on average over Europe. The decadal change in the number of hot days is also lower in TSMP
than on average in the CORDEX ensemble. TSMP systematically simulates fewer heat waves (i.e., heat events lasting 6 days or more) compared to the CORDEX ensemble, moreover, they are less intense. Southern Europe is particularly sensitive to groundwater coupling, while Scandinavia is the least sensitive. Therefore, an explicit representation of groundwater in RCMs may be a key in bias reduction in simulated duration and intensity of heat waves, especially in Southern Europe. The results emphasise the importance of groundwater coupling in long-term regional climate simulations and potential implications for
climate change projections.

# 1 Introduction

Over the past decades, the number of heat waves has increased (e.g., Frich et al., 2002; Alexander et al., 2006; Christidis et al., 2015; Zhang et al., 2020). The years 2003, 2010, 2018, and 2022 were exceptionally hot in Europe, characterised by record-breaking air temperatures (e.g., Stott et al., 2004; Barriopedro et al., 2011; Liu et al., 2020; Dirmeyer et al., 2021; Yule et al., 2023). With projected climate change, the occurrence of heat waves will continue to increase (e.g., Russo et al., 2015; Myhre et al., 2019; Hari et al., 2020; Molina et al., 2020; Masson-Delmotte et al., 2021), leading to multiple negative socio-economic impacts (e.g., Bosello et al., 2007; Ciscar et al., 2011; Amengual et al., 2014; Yin et al., 2022).

The underlying hydrometeorological mechanisms of heat waves have been extensively studied (e.g., Lhotka and Kyselý, 2015; Horton et al., 2016; Liu et al., 2020). Heat waves are triggered by strong, persistent quasi-stationary large-scale high pressure systems associated with atmospheric blocking events, resulting in subsiding, adiabatically warming air masses, and clear skies allowing for high insolation (Tomczyk and Bednorz, 2016; Horton et al., 2016; Kautz et al., 2022). Atmospheric blocking events also impact winter and early spring precipitation in most parts of Europe and, in turn, affect soil moisture (e.g., Vautard et al., 2007; Ionita et al., 2020). The evolution of heat waves depends primarily on the synoptic weather patterns in combination with ambient soil moisture conditions, further altered by multiple land-atmosphere feedback processes (e.g., Fischer et al., 2007; Horton et al., 2016).

Many European summer heat waves were preceded by a deficiency of spring precipitation (Dirmeyer et al., 2021; Stegehuis et al., 2021; Hartick et al., 2021). Due to the soil moisture memory effect, the lack of precipitation in early spring causes negative soil moisture anomalies in early summer and leads to strong land-atmosphere coupling (a measure of the response of the atmosphere to anomalies in the land surface state) with a lower evaporation fraction. This reduces latent cooling and amplifies summer temperatures (e.g., Fischer et al., 2007; Miralles et al., 2012; Knist et al., 2017; Dirmeyer et al., 2021). Note that soil moisture memory is a phenomenon of persistence of wet or dry anomalies over a long period of time, from weeks to months, after the atmospheric conditions that caused them have passed; this allows to preserve the hydroclimatic conditions of the preceding months (e.g., Manabe and Delworth, 1990; Song et al., 2019). Thus, depending on soil moisture conditions, the soil moisture memory effect can contribute to either buffering negative droughts impacts and weakening a heat wave, or, conversely, delaying drought recovery and exacerbating the occurrence of a heat wave (e.g., Erdenebat and Tomonori, 2018; Martínez-de la Torre and Miguez-Macho, 2019). In addition to precipitation, soil moisture is strongly influenced by groundwater dynamics via vertical fluxes across the water table (capillary rise) and via horizontal fluxes through gravity-driven lateral transport within the saturated zone. Here, the water table depth dictates the intensity of shallow groundwater–soil moisture and evapotranspiration coupling (Kollet and Maxwell, 2008).

In the context of climate impact assessments, dynamical downscaling of GCMs with RCMS is widely used to generate regional climate change scenario information (Vautard et al., 2013b; Mearns et al., 2015; Jacob et al., 2020). RCMs have been shown to provide added value to driving GCMs by better capturing small-scale processes (Giorgi and Gutowski, 2015; Torma et al., 2015; Prein et al., 2016; Iles et al., 2020; Rummukainen, 2016), but model biases (offset during the historical period against observations) and uncertainties in climate projections still remain (Hawkins and Sutton, 2009; Lhotka et al., 2018;

Sørland et al., 2018; Fernandez-Granja et al., 2021). In fact, many RCMs tend to overestimate the frequency, duration, and
intensity of heat waves (Vautard et al., 2013a; Plavcová and Kyselý, 2016; Lhotka et al., 2018; Furusho-Percot et al., 2022).

The role of soil moisture in modelling heat waves is crucial (e.g., Seneviratne et al., 2006, 2010; Fischer et al., 2007), but due
to the complexity of the feedbacks involved and related high computational cost, the explicit representation of hydrological pro-
cesses is oversimplified or neglected in most RCMs. Commonly applied hydrology schemes are based on 1D-parameterizations
in the vertical direction with runoff generation at the land surface and a gravity driven free drainage approach as the lower
boundary condition; in such a parametrisation there is no lateral subsurface flow and only the 1D-Richards' equation is solved
(e.g., Niu et al., 2007; Campoy et al., 2013). RCMs with a simplified representation of hydrological processes have difficulties
in reliably reproducing the land surface energy flux partitioning and, consequently, near-surface air temperatures, leading to
warm biases (Vautard et al., 2013a; Barlage et al., 2021; Furusho-Percot et al., 2022). Hydrological parameters tuning (e.g.,
Teuling et al., 2009; Bellprat et al., 2016) or developing new parameterizations of groundwater dynamics (e.g., Liang et al.,
2003; Yeh and Eltahir, 2005; Schlemmer et al., 2018) have been shown to improve model results. A physically consistent
description of hydrological processes in RCMs can be achieved by an explicit representation of 3D subsurface and ground-
water hydrodynamics together with overland flow. Thereby accounting for the feedback loops over the terrestrial system (e.g.,
Maxwell et al., 2007), i.e., the closure of water and energy cycles from groundwater across the land surface to the top of the
atmosphere, as for instance in the Terrestrial Systems Modelling Platform (TSMP) (Shrestha et al., 2014; Gasper et al., 2014),
a regional climate system model.

Keune et al. (2016) demonstrated the link between groundwater and near-surface air temperature in an analysis of the August
2003 European heat wave from TSMP simulations nested within ERA-Interim reanalysis (Dee et al., 2011). The model set up is
over the CORDEX European domain (Gutowski et al., 2016; Jacob et al., 2020) with two different groundwater configurations:
(i) simplified 1D free drainage approach and (ii) 3D physics-based variably saturated groundwater dynamics. The study clearly
showed the impact of groundwater dynamics on the land surface water and energy balance: latent heat fluxes were higher and
maximum temperatures were lower, especially in areas with shallow water table depth, in the 3D configuration compared to
the simplified 1D free drainage approach. Keune et al. (2016) suggest that the 3D groundwater dynamics in TSMP alleviate
the evolution of a single heat wave due to weaker land-atmosphere feedbacks compared to the simplified 1D free drainage
approach, at least during the investigated European heat wave of summer 2003.

Therefore, compared to the 1D approach, the 3D groundwater dynamics in TSMP lead to regionally shallow groundwater
levels, causing wetter soils, and a reduction in the Bowen ratio (i.e., ratio between sensible heat flux to latent heat flux) due to
an increase in surface latent heat flux and a decrease in surface sensible heat flux, that leads to increased evapotranspiration
(Maxwell and Condon, 2016). Such an increase in a latent heat flux also causes moistening of the lower atmosphere and in-
creases downward longwave radiation due to the greenhouse effect of water vapor, on the other hand, it cools the surface and
reduces outgoing surface longwave radiation (Pal and Eltahir, 2001). In addition, increased evapotranspiration may cause moist
convection or rainfall, which further affects soil moisture (Eltahir, 1998; Yang et al., 2018). In its turn, the simplified repre-
sentation of groundwater dynamics with the 1D free drainage approach leads to the opposite effect, namely an overestimation
of the land surface-atmosphere coupling, i.e., deeper groundwater levels cause drier soils, an increase in the Bowen ratio, a

decrease in cloud cover and enhancement of net solar radiation and a reduction in downward longwave radiation (Hartick et al., 2022), and, as a result, higher near-surface temperatures, which in turn further reduces soil moisture (Vogel et al., 2018). The ability of groundwater to decrease warm summer biases and moderate maximum air temperatures during a single seasonal heat wave in RCM simulations was also discussed in Barlage et al. (2015, 2021) and Mu et al. (2022).

Further studies were carried out to understand whether the observed differences in simulated near-surface temperature due to differences in groundwater configuration persist over a long time period, and how this manifests itself for heat waves in the EURO-CORDEX realm. Furusho-Percot et al. (2019) showed that TSMP evaluation run (1996–2018) forced by the ERA-Interim reanalysis is able to capture climate system dynamics and the succession of warm and cold seasons on the regional scale for the PRUDENCE regions of Europe (Christensen and Christensen, 2007) consistently with E-OBS observations (Cornes et al., 2018). Furusho-Percot et al. (2022) demonstrated that TSMP multiannual simulations exhibit lower deviations of summer heat wave indices from the E-OBS observational dataset, compared to ERA-Interim driven RCM evaluation simulations of the EURO-CORDEX experiment, which tend to simulate too persistent heat waves. This particular behaviour of TSMP is attributed to its improved hydrology. The improved capacity to sustain soil moisture translates into more reliable latent heat flux and evapotranspiration, that, in turn, leads to a decrease in the heat wave intensity, spatial extent, and the number of days with anomalously high near-surface temperatures. An important question still remains: how will these findings be reflected in long-term regional climate simulations?

In this paper, we present a unique dataset from TSMP forced by the Max Planck Institute Earth System Model at Low Resolution, MPI-ESM-LR (Giorgetta et al., 2013), historical boundary conditions in the context of EURO-CORDEX GCM-RCM downscaling and long-term climate modelling. We interrogate the statistics of heat events (i.e., a series of consecutive days with a near-surface temperature exceeding the 90th percentile) characteristics (frequency, duration, intensity) for the summers of 1976-2005 with respect to the reference period 1961-1990 by comparing TSMP results with the EURO-CORDEX multi-model RCM ensemble driven by GCM control simulations of phase five of the Coupled Model Intercomparison Project (CMIP5) (Taylor et al., 2012), to understand the influence of 3D groundwater dynamics on simulated heat events for historical regional climate simulations and potential consequences for ensuing climate change projections. While the 1996-2018 TSMP evaluation runs nested within ERA-Interim reanalysis were examined for heat wave statistics (Furusho-Percot et al., 2022), long-term historical climate simulations of TSMP forced by GCM have not been previously presented. Thus, this is the first downscaled regional historical climate simulation from groundwater across the land surface to the top of the atmosphere placed in the context of the climate scenario runs of the EURO-CORDEX RCM ensemble and analysed for summer heat events.

In Sec. 2, we describe TSMP setup and configuration, the ensemble of the EURO-CORDEX climate change scenario GCM-RCM control runs, and the methodology of heat events analysis. In Sect. 3, we examine the new GCM-RCM TSMP-MPI dataset for consistency with the CORDEX ensemble, and we present the results on the impact of 3D groundwater dynamics on simulated heat events for regional historical climate simulations. Section 4 provides a summary and overall conclusions.

## 2 Methods

### 2.1 TSMP

TSMP is a scale-consistent, highly modular, fully integrated soil-vegetation-atmosphere modelling system (e.g., Shrestha et al., 2014; Gasper et al., 2014). TSMP consists of three component models: the atmospheric COnsortium for Small Scale Modelling (COSMO) model version 5.01 (e.g., Baldauf et al., 2011), the Community Land Model (CLM) version 3.5 (e.g., Oleson et al., 2004, 2008), and the hydrological model ParFlow version 3.2 (e.g., Maxwell and Miller, 2005; Kollet and Maxwell, 2006; Kuffour et al., 2020). The component models are externally coupled via the Ocean Atmosphere Sea Ice Soil (OASIS) version 3.0 Model Coupling Toolkit (MCT) coupler (e.g., Valcke, 2013), which enables closure of the terrestrial water and energy cycles from the bedrock to the top of the atmosphere. For details on TSMP, see Shrestha et al. (2014); Gasper et al. (2014).

COSMO is a non-hydrostatic limited-area atmospheric model. It is based on the primitive thermo-hydrodynamical Euler equations formulated in rotated geographical coordinates and generalized terrain-following height coordinates, describing compressible flow in a moist atmosphere. COSMO parameterization schemes cover various physical processes, such as radiation, cloud microphysics, deep convection, etc. The boundary conditions used for COSMO are typically provided by a coarse grid model. In the coupled setup of TSMP, the lower boundary conditions for COSMO (e.g., surface albedo, energy fluxes, surface temperature, surface humidity) are provided by CLM.

CLM is a biogeophysical model of the land surface. It simulates land-atmosphere exchanges in response to atmospheric forcings. CLM consist of four components that describe biogeophysics, hydrologic cycle, biogeochemistry, and dynamic vegetation. In TSMP, CLM receives short-wave radiation, wind speeds, barometric pressure, precipitation, near-surface temperature, and specific humidity from COSMO. In turn, CLM sends to ParFlow infiltration and evapotranspiration fluxes for each soil layer.

ParFlow is a hydrological model that simulates variably saturated three-dimensional subsurface hydrodynamics using Richards equation integrated with shallow overland flow based on a kinematic wave approximation. ParFlow allows 3D-redistribution of subsurface water in a continuum approach. In TSMP, ParFlow replaces the hydrologic functionality of CLM.

### 2.2 Model setup

TSMP simulations in this study are conducted for the historical time period from December 1949 to the end of 2005 over the European continent according to the EURO-CORDEX simulation protocol (e.g., Gutowski et al., 2016) using rotated latitude-longitude model grid with a horizontal resolution of $0.11°$ (EUR-11) or about $12.5\,\mathrm{km}$. Note, the simulations represent the first EURO-CORDEX climate change control simulations with explicit representation of 3D groundwater. COSMO extends vertically to $22\,\mathrm{km}$ subdivided into 50 levels. The COSMO configuration used in the TSMP setup resembles that of the COSMO model in CLimate Mode (CCLM) (e.g., Rockel et al., 2008). CLM has 10 soil layers with a total depth of $3\,\mathrm{m}$. These layers coincide with the 10 top layers of ParFlow, which has 5 additional layers that increase in thickness to a total depth of $57\,\mathrm{m}$.

The time step for ParFlow and CLM is 900 sec, for COSMO it is 75 sec. The coupling time step between TSMP component models is 900 sec.

For CLM, plant functional types (PFT) are taken from the Moderate Resolution Imaging Spectroradiometer (MODIS) dataset (Friedl et al., 2002). Leaf area index, stem area index, and the monthly bottom and top heights of each PFT are calculated based on the global CLM surface dataset (Oleson et al., 2008). Compared to the previous studies of Furusho-Percot et al. (2019, 2022); Hartick et al. (2021), where soil parameters were assumed to be vertically homogenous in ParFlow, in this work we have improved the subsurface hydrogeology, which is described below. Static input fields (i.e., soil color, percentage clay, percentage sand, dominant land use type, dominant soil types in the top layers, dominant soil types in the bottom layers and subsurface aquifer and bedrock bottom layers) are derived from MODIS, Food and Agriculture Organization soil database (FAO, 1988), pan-European River and Catchment Database (Vogt et al., 2007), International Hydrogeological map of Europe (IHME) (Duscher et al., 2015) and the GLobal HYdrogeology MaPS (GLHYMPS) (Gleeson et al., 2014). In particular, the bedrock geology is constructed using the IHME dataset and the lower resolution GLHYMPS. The pan-European River and Catchment Database serves in ParFlow as a proxy for the alluvial aquifer system, with the assumption that alluvial aquifers lie underneath or in proximity of existing rivers.

Forcing data for the TSMP atmospheric component model, i.e., for COSMO, are provided by the Max-Planck Institute's MPI-ESM-LR r1i1p1 CMIP5 GCM experiment, with a resolution of T63L47 (Giorgetta et al., 2013). CLM and ParFlow are initialised (i.e., land surface, subsurface hydrology, and energy states) with the moisture conditions of the 1st of December 2011 from the previous evaluation run driven by ERA-Interim reanalysis (Furusho-Percot et al., 2019). In the analysis, we discard the first 10 years of TSMP simulations due to hydrodynamic spin-up.

## 2.3 EURO-CORDEX ensemble

The selected EURO-CORDEX ensemble members of the multi-physics RCM climate change scenario control runs driven by different CMIP5 GCMs (r1i1p1 ensemble members) on 0.11° grid (EUR-11) is used in conjunction with the coupled TSMP modelling platform to study the characteristics of summer heat events. Note that CMIP5 GCM historical control simulations are performed under observed natural and anthropogenic forcing (Taylor et al., 2012). Based on availability, the following EURO-CORDEX simulations are considered, identified by their institution, and the RCM and GCM IDs (Table 1): CCLM4-8-17 (CCLM4-8-17 forced by MPI-ESM-LR and CNRM-CM5), CLMcom-ETH (COSMO-crCLIM forced by MPI-ESM-LR, CNRM-CM5, and NCC-NorESM1-M), MPI-CSC (REMO2009 driven by MPI-ESM-LR), GERICS (REMO2015 forced by NCC-NorESM1-M, NOAA-GFDL-ESM2G, and IPSL-CM5A-LR). The considered CORDEX multi-model ensemble includes two main groups of RCMs, namely COSMO (v5.01 in TSMP, v4.8, and an accelerated version of COSMO in COSMO-crCLIM) and REMO (v2009 and v2015), driven by 5 different GCMs, for a total of 10 different GCM-RCM pairs. For a reliable analysis, Giorgi and Coppola (2010) points out that a subset of at least 4-6 models is needed, and Déqué et al. (2007) shows that the number of GCMs involved should at least be the same as the number of RCMs. The CORDEX RCM most compatible with TSMP is CCLM4-8-17, where the largest differences arise from the lower boundary condition in COSMO. In TSMP, the lower boundary condition for COSMO accounts for groundwater feedbacks due to the coupling between the land

**Table 1.** The ensemble of EURO-CORDEX climate change scenario control runs.

| GCM-RCM | MPI-ESM-LR *(Mauritsen et al., 2019)* | CNRM-CM5 *(Voldoire et al., 2013)* | NCC-NORESM1-M *(Bentsen et al., 2013)* | NOAA-GFDL-ESM2G *(Dunne et al., 2012)* | IPSL-CM5A-LR *(Dufresne et al., 2013)* |
|---|---|---|---|---|---|
| TSMP *(Shrestha et al., 2014)* | X | | | | |
| CCLM4-8-17 *(Rockel et al., 2008)* | X | X | | | |
| COSMO-crCLIM *(Pothapakula et al., 2020)* | X | X | X | | |
| REMO2009 *(Jacob and Podzun, 1997)* | X | | | | |
| REMO2015 | | | X | X | X |

surface model CLM and the hydrologic model ParFlow, unlike in the CCLM where the soil processes are modelled with the TERRA-ML soil-vegetation land surface model (Grasselt et al., 2008; Doms et al., 2013). All members of the ensemble, except

for TSMP, include simplified representations of subsurface hydrodynamics.

Note that the ensemble of EURO-CORDEX climate change scenario RCM control runs is not intended for direct comparison between individual models, as it includes different RCMs in combination with different driving GCMs. Therefore, due to connections of various factors (e.g., model setups, conceptual and structural model uncertainties, different physical parameterizations, internal variability, representation of subsurface-land-atmosphere interactions, lower and lateral atmospheric GCM

boundary conditions, etc.) in addition to the groundwater coupling, it is challenging in the multi-model CORDEX ensemble to reveal the exact cause and effect relationships of the explicit groundwater representation for simulated hot days and associated characteristics of heat events in RCMs. However, the consideration of an extended period, e.g., 30 years, allows to draw statistical conclusions. In this study, the aim of the analysis of the TSMP historic simulations in the context of the CORDEX RCM ensemble is to interrogate whether the new TSMP driven by MPI-ESM-LR GCM-RCM dataset is consistent with the

CORDEX ensemble and, in particular, to gain insight into the role of groundwater for long-term climate simulations from the statistical analysis of heat events.

## 2.4 Analysis of heat events

There is no universally accepted method for defining heat events, but the most commonly used approach is built on a percentile temperature threshold (e.g., Zhang et al., 2005, 2011; Sulikowska and Wypych, 2020). Note that although the focus

is on temperature-based diagnostics, it is often ambiguous or inconsistent, describing heat events only partially (Perkins and Alexander, 2013).

In this study, we define a hot day as a day with a daily mean temperature above the local 90th percentile from the reference 1961-1990 period. We calculate the 90th percentile for each EUR-11 grid point of the EURO-CORDEX domain for every

summer day from a consecutive 5-day moving window centered on that calendar day from the 30-year reference period between 1961 and 1990. The first occurrence of a hot day determines the start of a heat event. A series of hot days constitutes a

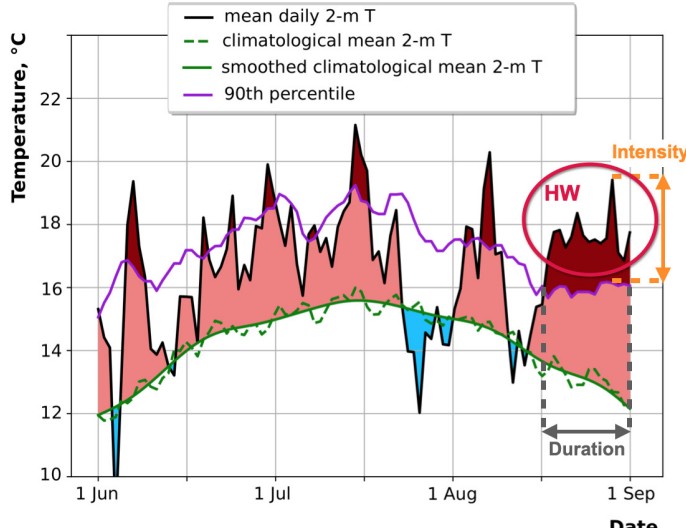

| Heat event # | Start date | End date | Duration, days | Intensity, °C |
|:---:|:---:|:---:|:---:|:---:|
| 1. | 1.06 | 1.06 | 1 | 0,26 |
| 2. | 6.06 | 9.06 | 4 | 2,71 |
| 3. | 19.06 | 19.06 | 1 | 1,30 |
| 4. | 24.06 | 24.06 | 1 | 1,07 |
| 5. | 30.06 | 1.07 | 2 | 1,40 |
| 6. | 7.07 | 7.08 | 2 | 0,41 |
| 7. | 15.07 | 17.07 | 3 | 1,90 |
| 8. | 3.08 | 4.08 | 2 | 0,63 |
| 9. | 6.08 | 7.08 | 2 | 3,08 |
| **10.** | **17.08** | **31.08** | **15** | **3,28** |

**Figure 1.** Schematic of summer heat wave (HW) detection. An example is given for June-July-August of 1972 for one grid point [250, 300] of the EURO-CORDEX domain. Data taken from the TSMP simulations. The solid black line is the mean daily 2 m air temperature. The dashed green line represents the climatological (1961-1990) mean daily 2 m air temperature, and the solid green line represents the same dependence with a Butterworth filter. The solid violet line is the 90th percentile of the mean daily 2 m air temperature in summer calculated from a 5-day window centered on each calendar day for the 1961-1990 reference period. The shaded light red colour indicates days with temperatures above the climatological mean, and the shaded dark red colour emphasizes days with temperatures above the 90th percentile, i.e. "hot days", "heat events", or "heat waves".

heat event, highlighted in dark red in Fig. 1. A heat event is interrupted if the mean daily temperature drops below the 90th percentile-based threshold. The total number of hot days during the investigated period corresponds to the TG90p heat index from the joint CCl/CLIVAR/JCOMM Expert Team on Climate Change Detection and Indices (ETCCDI) (e.g., Karl et al., 1999). TG90p describes the number of days with $TG_{ij} > TG_{in}90$, where $TG_{ij}$ is the mean daily temperature on day $i$ of period $j$ and $TG_{in}90$ is the $i$-day 90th percentile calculated from a 30-year reference period $n$.

A heat event can be characterised by its duration, intensity, and frequency (e.g., Horton et al., 2016). A heat event duration is the number of consecutive days over which the heat event lasts. If a heat event lasts long enough, it can be classified as a heat wave. Similar to Fischer and Schär (2010), we consider a heat wave as a spell of at least six consecutive days with mean daily temperatures above the local 90th percentile of the reference 1961-1990 period. See Fig. 1 for an explanation of the heat wave detection. Therefore, we consistently use the terminology "hot day", "heat event", and "heat wave" throughout this analysis.

A heat event intensity is the maximum of the difference between the mean daily temperature and the 90th percentile of the reference 1961-1990 period within a single heat event (e.g., Vautard et al., 2013a). Intensity represents the severity of a heat event (see Fig. 1). Adopting the definition from the heat wave duration index (e.g., Frich et al., 2002; Espírito Santo et al.,

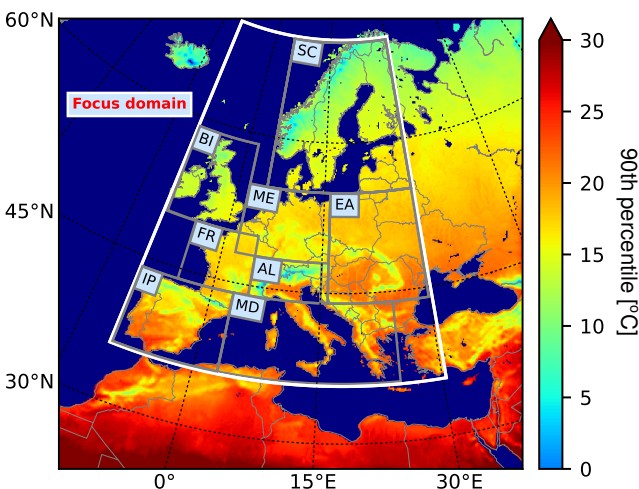

**Figure 2.** Mean of the 90th percentile of 2 m air temperatures in summer simulated with TSMP. The 90th percentile is calculated from a consecutive 5-day moving window centered on each calendar day of the summer season from the 30-year reference period from 1961 to 1990. The white box indicates the focus domain for the analysis in our study [10°W-30°E, 36°N-70°N]. PRUDENCE analysis regions are shown as grey boxes: British Isles (BI), Iberian Peninsula (IP), France (FR), Mid-Europe (ME), Scandinavia (SC), Alps (AL), Mediterranean (MD) and Eastern Europe (EA).

2014), we classify a heat wave as intense if it exceeds 5 K. Some studies also classify heat waves according to their intensity as low, severe, or extreme (e.g., Nairn and Fawcett, 2014).

A frequency of heat events of a certain type (e.g., specific duration or intensity) during the investigated period is the number of specific heat events divided by the total number of all heat events that occurred during this investigated period (e.g., Vautard et al., 2013a). For example, in Fig. 1, the frequency of heat events with 2 days duration is the number of those heat events
(i.e., 4) divided by the total number of all heat events (i.e., 10). The resulting frequency is 0.4 and indicates that 40% of all heat events have a duration of 2 days.

We examine heat events in Europe by assessing their characteristics explained above based on mean daily 2 m air temperatures on the native EUR-11 grid in the ensemble of EURO-CORDEX climate change scenario RCM control runs driven by different GCMs, as listed in Table 1. The analysis is performed for the summer season of the 30-year period, from 1976 to 2005
with regard to the reference period from 1961 to 1990 in each RCM. The analysis is conducted over the focus domain covering the European continent [10°W-30°E, 36°N-70°N] as shown in Fig. 2. Note that we analyse only land grid elements.

## 3   Results

### 3.1   Hot days number

In order to evaluate the impact of groundwater coupling on the interannual variability of hot days during the summer season in RCMs, we examine the occurrence of hot days in the focus domain in the ensemble of the EURO-CORDEX climate change

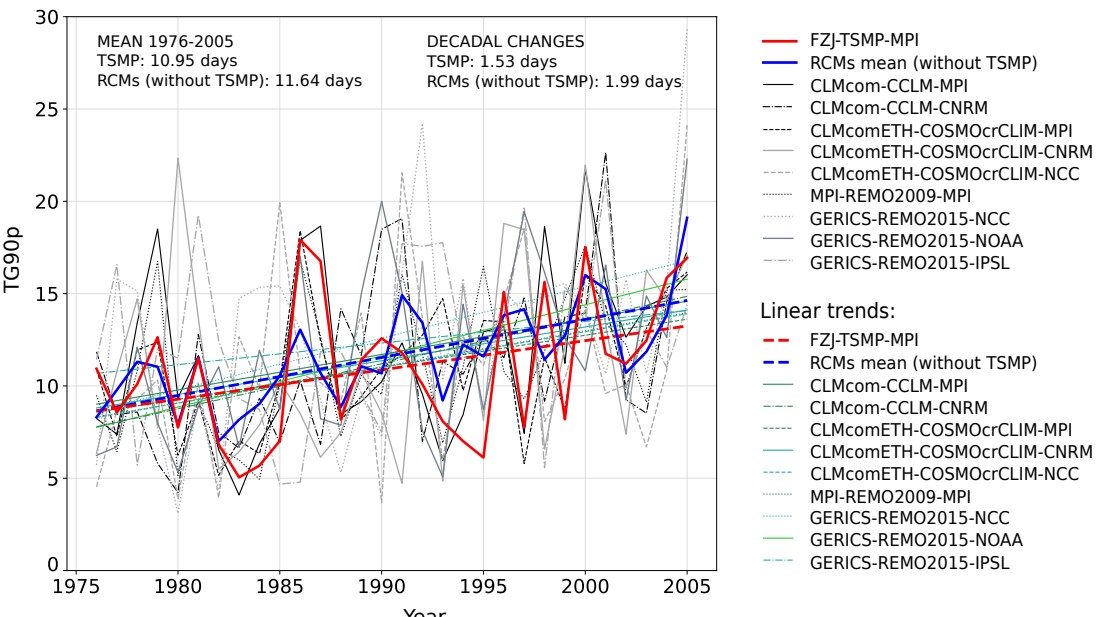

**Figure 3.** Time series of the mean hot days number (TG90p index) during the summer season over the focus domain and its linear trends during 1976-2005 with respect to the reference period 1961-1990, in the ensemble of EURO-CORDEX climate change scenario RCM control runs. Averaging of TG90p is performed over the total number of land grid points in the focus domain. The solid and dashed red lines show the TG90p mean from the TSMP simulations, as well as its linear trend. The black and grey lines represent the TG90p mean from the CORDEX ensemble and the different green lines are their linear trends respectively. The TG90p averaged over the multi-model RCM CORDEX ensemble (excluding TSMP) is shown with the solid blue line and its linear trend is shown with the dashed blue line.

scenario GCM-RCM historical control runs from 1976 to 2005 with regard to the reference period 1961-1990. A comparison of the mean hot days number (i.e., TG90p index) per summer over the focus domain from TSMP and the CORDEX ensemble suggests that the impact of groundwater coupling varies from year to year (Fig. 3). Here, the long-term soil moisture memory effects can play an important role, for example by increasing the probability of a water deficit in regions that had a water deficit in the previous year (e.g., Hartick et al., 2021), and consequently influencing the hot days occurrence. A positive linear trend in the TG90p index is observed in all considered RCMs on average in the focus domain (see Fig. 3). The decadal change in the TG90p index in TSMP is 1.53 days. In contrast, the decadal change in the TG90p index averaged over the multi-model RCM CORDEX ensemble (excluding TSMP) is higher, up to 2 days.

A spatial distribution of the TG90p mean and TG90p variability, as well as the decadal change of TG90p is shown in Fig. 4-6. Uncertainty in simulated near-surface temperature in summer is strongly controlled by the large-scale atmospheric circulation imposed by the boundary conditions (e.g., Déqué et al., 2007; Fernández et al., 2019), with larger impact in the South-West (IP, FR) than the North-East (SC, ME, EA) PRUDENCE regions (Déqué et al., 2012). Hence the spatial pattern of the TG90p index significantly differs between different GCM-RCMs and the results from RCMs driven by the same GCMs show a rather similar behaviour. TSMP produces the smoothest spatial distribution of the TG90p mean and TG90p variability

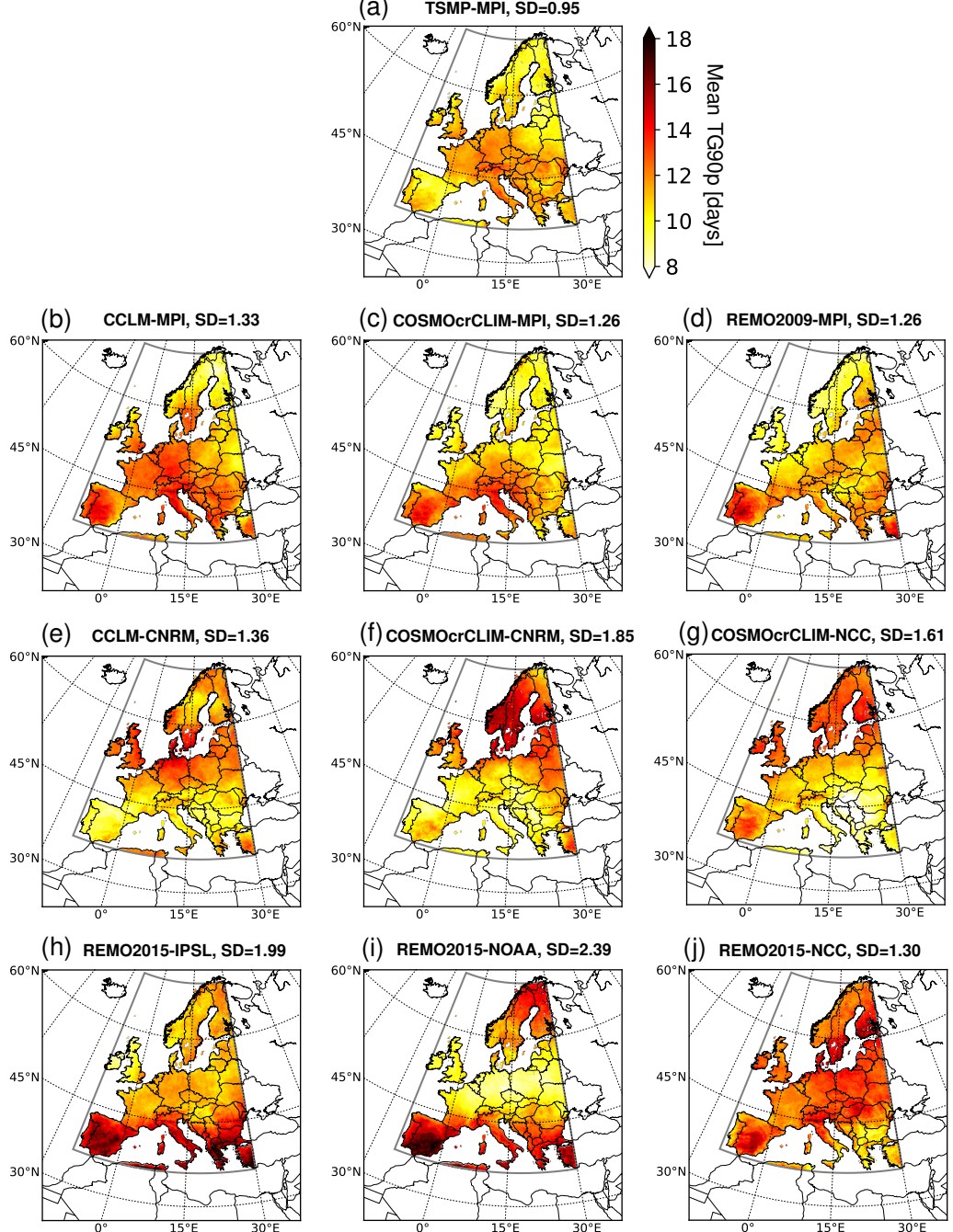

**Figure 4.** Spatial distribution of the number of hot days (TG90p index) averaged between 1976 and 2005 with respect to the reference period 1961-1990 for the summer season, in the ensemble of EURO-CORDEX climate change scenario RCM control runs. The standard deviation (SD) of the spatial distribution is also indicated on each figure.

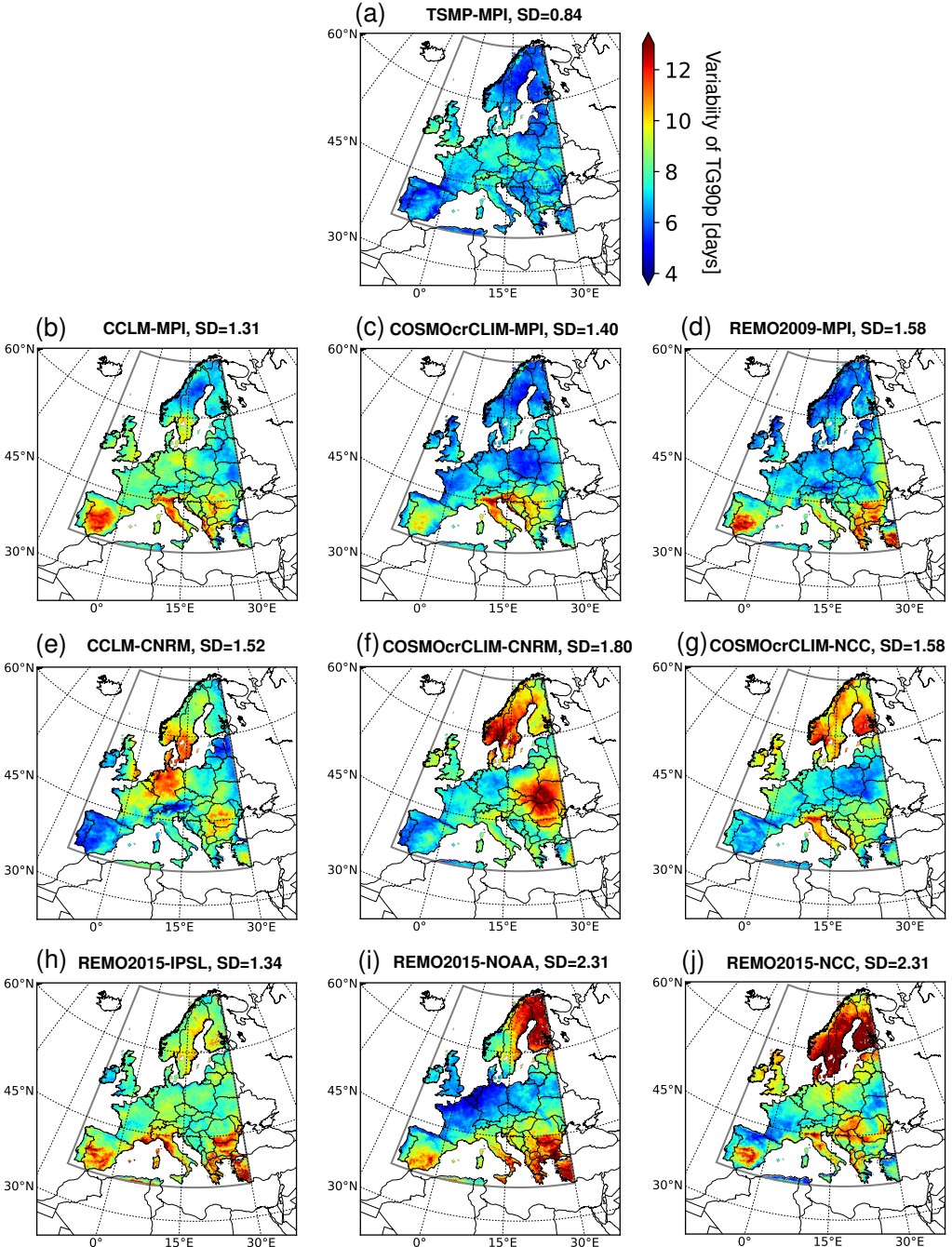

**Figure 5.** Variability of the hot days number (TG90p index) for the summer season, calculated for each land grid element as the standard deviation of TG90p between 1976 and 2005, in the ensemble of EURO-CORDEX climate change scenario RCM control runs. The standard deviation (SD) of the spatial distribution is also indicated on each figure.

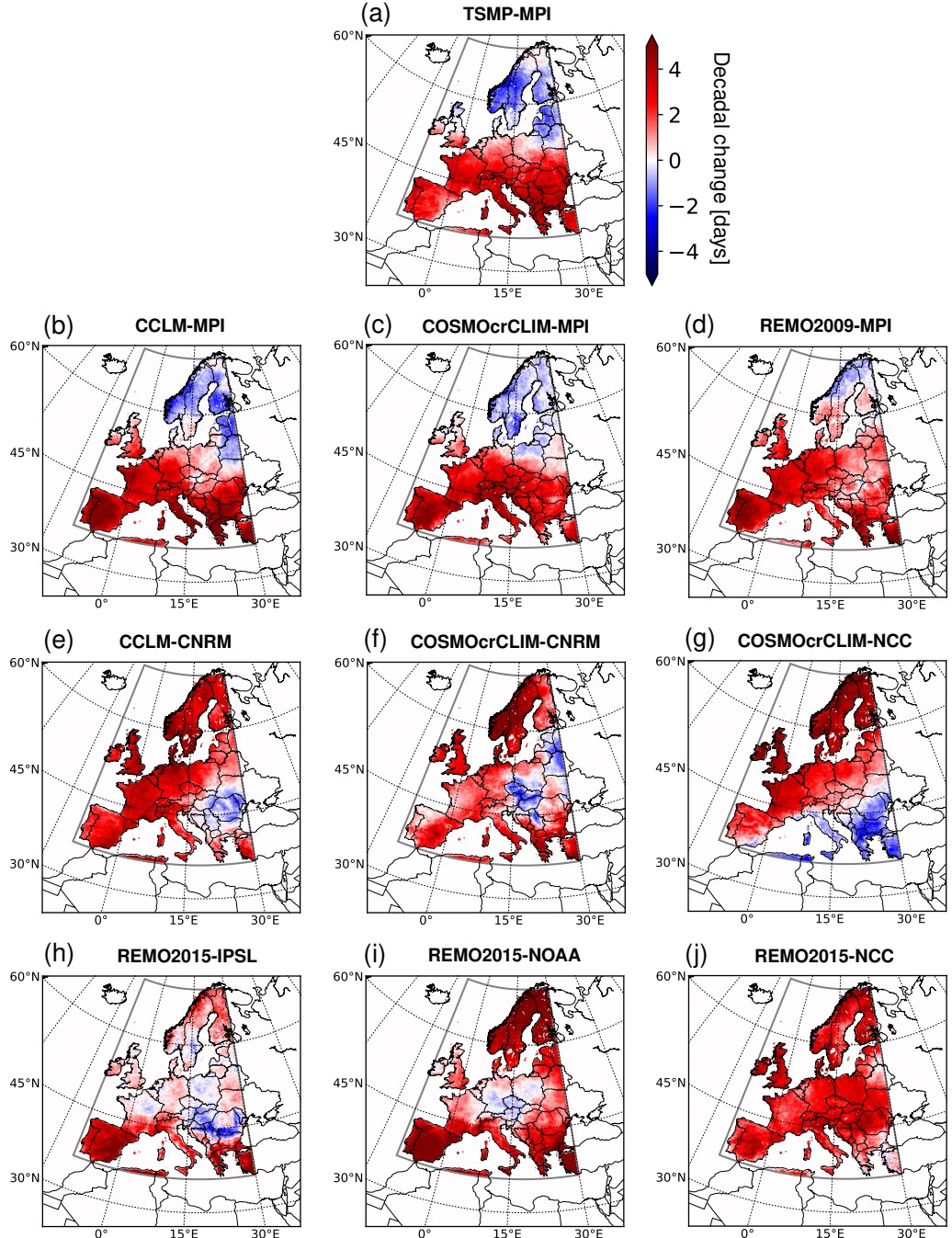

**Figure 6.** Spatial distribution of the decadal change in the number of hot summer days (TG90p index), based on data from 1976 to 2005 with respect to the reference period 1961-1990, in the ensemble of EURO-CORDEX climate change scenario RCM control runs. Decadal change is calculated as a linear trend for every land grid element.

compared to the CORDEX ensemble (see the standard deviation indicated in Fig. 4, 5). Thus, in TSMP, the regional difference in the distribution of the hot days number is smaller, and the climate is more steady with respect to the interannual changes in the simulated number of hot days in summer. In the considered ensemble, the TG90p mean and TG90p variability averaged over the focus domain are lowest in TSMP driven by MPI-ESM-LR, 10.95 days and 6.80 days, respectively, and highest in REMO2015 driven by NCC-NorESM1-M, 12.72 days and 9.42 days, see Fig. A1-A2 in Appendix A for details. The decadal change in the TG90p index averaged over the focus domain ranges from the lowest value of 1.13 days in REMO2015 driven by IPSL-CM5A-LR and the highest of 2.68 days in REMO2015 driven by NOAA GCM (Fig. A3 in Appendix A).

In particular, the TG90p index simulated by TMSP is consistent with the CORDEX RCMs driven by the same MPI-ESM-LR GCM, although there are some regional differences (see Fig. 4a-d, Fig. 5a-d). The largest differences are found in the Iberian Peninsula, where TSMP produces the TG90p mean of 10.36 days and the TG90p mean simulated by the CORDEX MPI-ESM-LR driven RCMs is 12.54-12.75 days (see Fig. A1 in Appendix A). At the same time, the interannual TG90p variability in the Iberian Peninsula reaches 6.17 days in TSMP and 8.01-9.59 days in the CORDEX MPI-ESM-LR driven RCMs (see Fig. A2 in Appendix A). All considered RCMs driven by MPI-ESM-LR simulate a negative decadal change in the TG90p index in Scandinavia, and a positive decadal change is observed in Southern and Central Europe (Fig. 6a-d). TSMP simulates, on average, a lower decadal change in the TG90p index in the focus domain than the CORDEX RCMs driven by MPI-ESM-LR (see Fig. A3 in Appendix A). The largest differences in the decadal change of the TG90p appear over the Iberian Peninsula, where TSMP produces 2.26 and CORDEX RCMs driven by MPI-ESM-LR simulate an increase of 3.66-4.25 hot days per decade.

From a comparison of TSMP and its most compatible CORDEX RCM, i.e., CCLM driven by MPI-ESM-LR, with the largest differences in the lower boundary condition for COSMO accounting for groundwater feedbacks in TSMP, TSMP simulates overall lower TG90p mean and TG90p variability in all PRUDENCE regions, with the largest discrepancies over the Iberian Peninsula and the Mediterranean and the smallest differences in Scandinavia. The decadal change in the TG90p is also lower in all PRUDENCE regions except the Alps and Eastern Europe. Different responses to groundwater coupling in different PRUDENCE regions may be explained by the soil moisture-temperature feedback associated with different evaporative regimes, energy-limited in Scandinavia and Northern Europe versus moisture-limited in Southern Europe (e.g., Koster et al., 2009; Seneviratne et al., 2010; Jach et al., 2022).

## 3.2 Heat events of different durations

The average summer seasonal number of heat events (i.e., a series of consecutive hot days) of different duration that occur in the focus domain between 1976 and 2005 is presented in Fig. 7a. The total number of heat events per land grid element of the focus domain varies between 4.18 in COSMO-crCLIM driven by CNRM-CM5 and 4.86 in REMO2015 driven by NCC-NorESM1-M. The ratio of the number of heat events between CORDEX RCMs and TSMP (blue lines in Fig. 7a) increases towards heat events of long durations ($\geq$6 days), i.e., heat waves, and indicates that TSMP systematically simulates the least number of heat waves compared to the CORDEX ensemble. REMO RCMs tend to simulate more heat waves with long durations than COSMO RCMs.

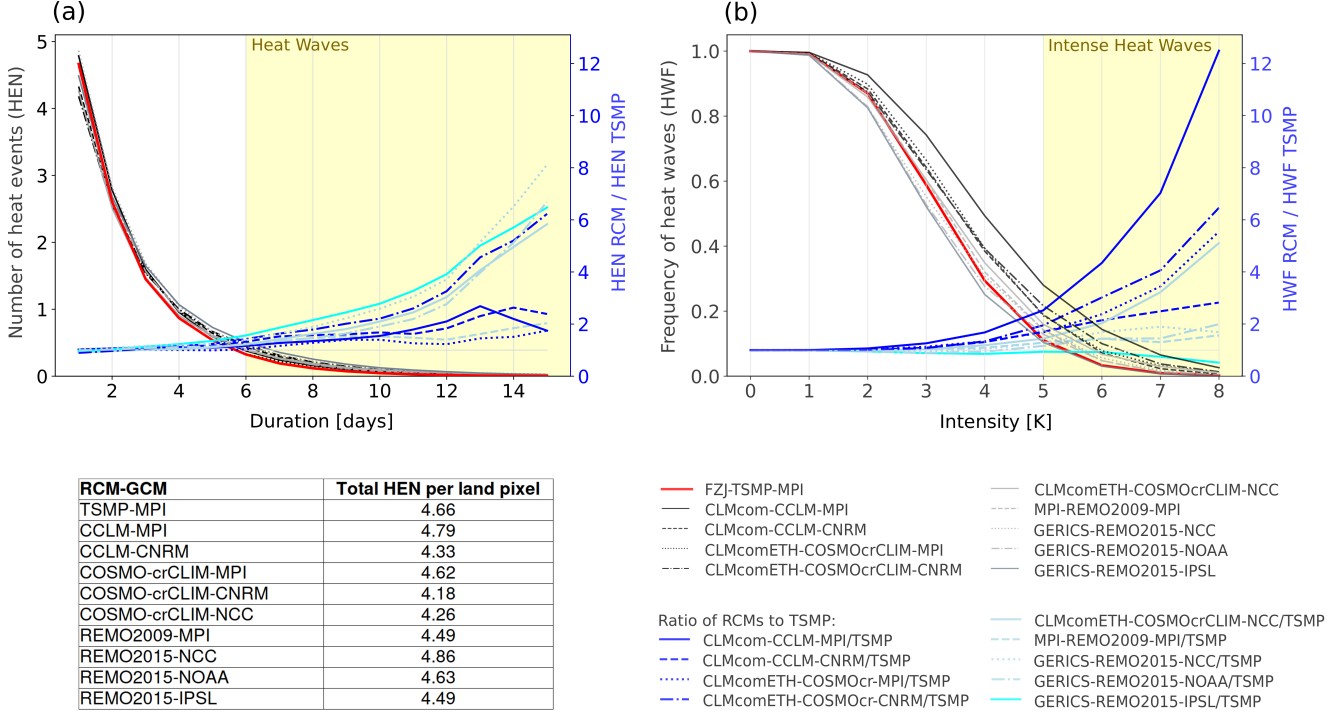

**Figure 7.** (a) Average number of heat events (HEN, y-axis) of duration equal or larger than a given number of days (x-axis) as a function of this number of days; the averaging is performed over the total number of land grid elements of the focus domain and 30 years, from 1976 to 2005. (b) Frequency of heat waves (HWF, y-axis) with an intensity higher or equal than a given value in abscissa occurring in the focus domain from 1976 to 2005 as a function of this intensity. The panels also show the ratio of HEN and HWF from RCMs and TSMP. Data are taken from the summer seasons between 1976 and 2005 with respect to the reference period 1961-1990 in each RCM of the CORDEX ensemble. The representation of the dependencies is adopted from the work of Vautard et al. (2013a).

Different GCM-RCMs simulate different spatial distributions of heat waves, whereas TSMP produces again the smoothest distribution of the decadal number of heat waves compared to the CORDEX ensemble, resulting in the least regional differences (Fig. 8). The number of heat waves over a decade in the focus domain ranges from the lowest value of 3.25 in TSMP driven by MPI-ESM-LR, to the highest of 5.09 in REMO2015 driven by IPSL-CM5A-LR (Fig. B1 in Appendix B). Comparing TSMP with CORDEX MPI-driven RCMs, CORDEX RCMs driven by MPI-ESM-LR simulate the highest number of heat waves toward Southern Europe, while in TSMP the most heat waves are located in Central Europe (see Fig. 8a-d). Strong differences between TSMP and the CORDEX MPI-driven RCMs appear over the Iberian Peninsula and the Mediterranean, and the smallest differences are in Scandinavia (see Fig. B1 in Appendix B). TSMP simulates fewer heat waves in all PRUDENCE regions except Mid-Europe, compared to the most compatible available CORDEX RCM, CCLM forced by MPI-ESM-LR.

The contribution of heat waves to the total number of hot days during the summer season varies among GCM-RCMs (Fig. 9). Heat waves account from 22.38 % of hot days in TSMP driven by MPI-ESM-LR to 34.40 % in REMO2015 driven by IPSL-CM5A-LR, on average in the focus domain (Fig. B2 in Appendix B). Therefore, the proportion of heat events that do not belong to heat waves is higher in TSMP compared to the CORDEX ensemble, indicating that TSMP generates more heat events with

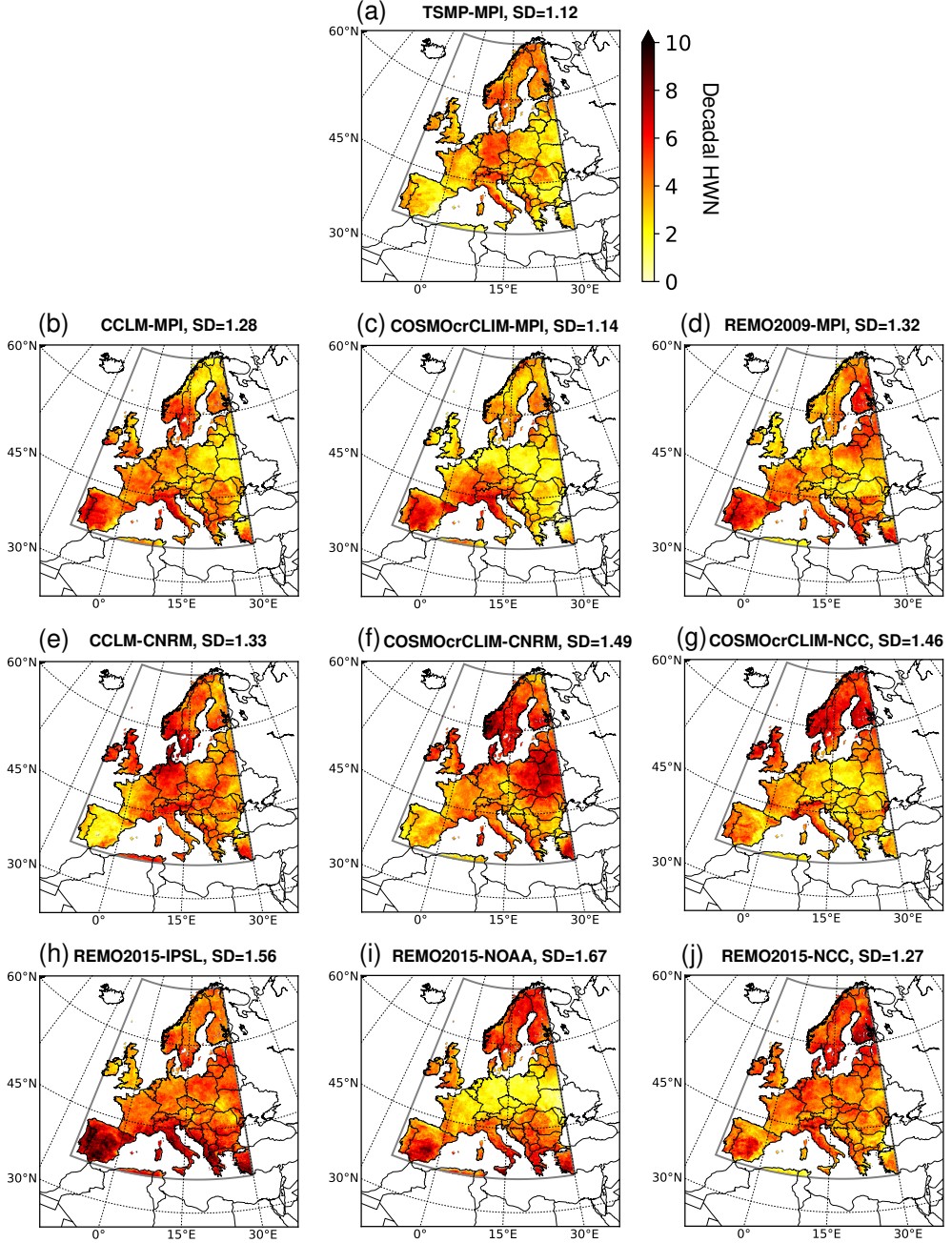

**Figure 8.** Spatial distribution of the heat waves number (HWN) over a decade based on data from 1976 to 2005 with respect to the reference period 1961-1990, in the ensemble of EURO-CORDEX climate change scenario RCM control runs (see Table 1). The standard deviation is also indicated above each figure.

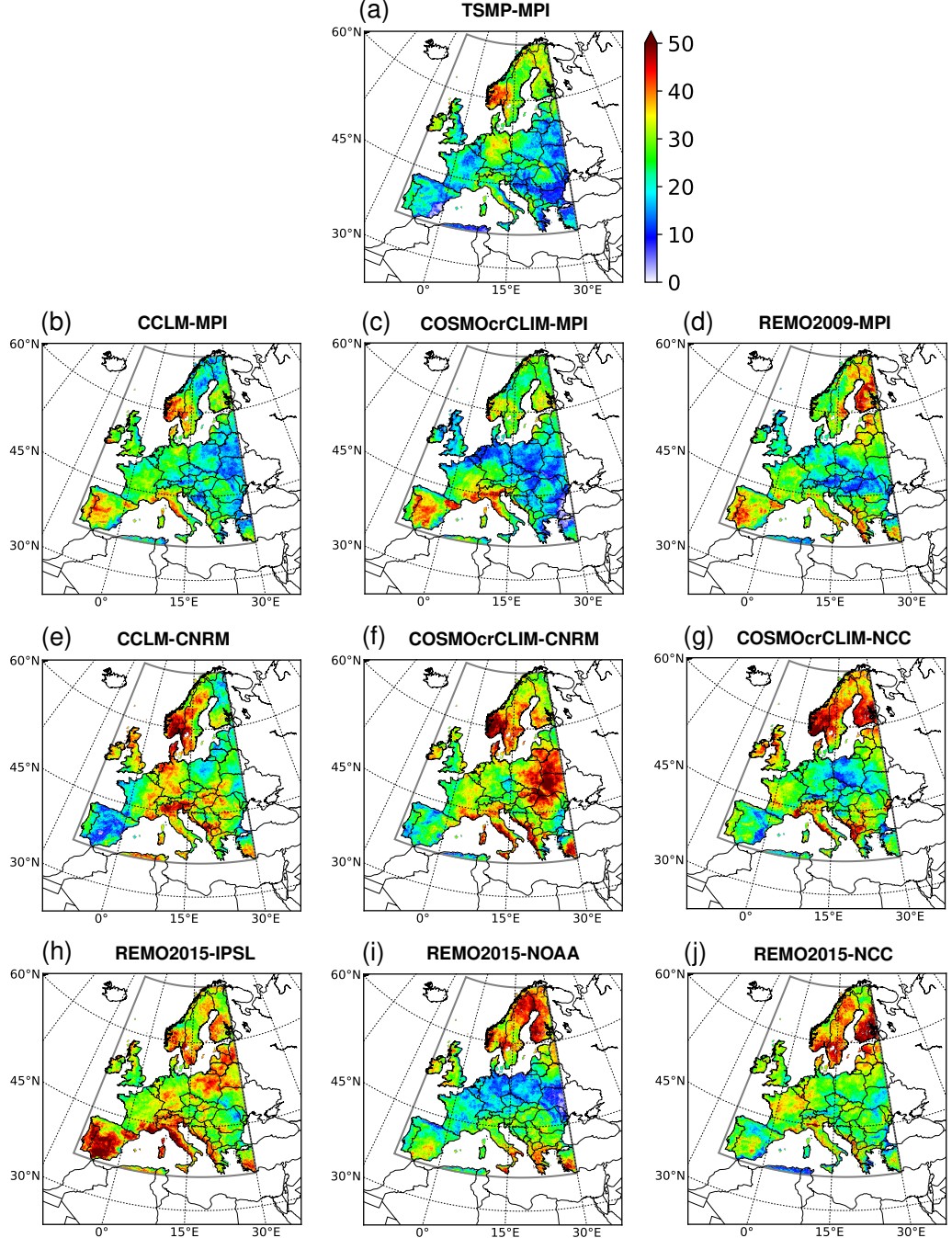

**Figure 9.** Contribution of heat waves to the number of hot days [%], calculated from the number of hot days and heat waves accumulated between 1976 and 2005, in the ensemble of EURO-CORDEX climate change scenario RCM control runs.

a duration of fewer than 6 days. Scandinavia is the region with the largest contribution of heat waves to the total number of hot days, on average in the considered RCM ensemble, which is expected to coincide with the region with the highest number of heat waves (see Fig. 8). Eastern Europe is the region with the least number of heat waves and the smallest contribution of heat waves to the total number of hot days. The largest discrepancy between TSMP and the CORDEX RCMs driven by the MPI-ESM-LR appear in the Iberian Peninsula and the Mediterranean.

### 3.3 Heat waves of different intensities

The dependence of the frequency of heat waves, which occurred between 1976 and 2005 in the focus domain, on the intensity is shown in Fig. 7b. The maximum frequency of heat waves is equal to 1 for an intensity greater than 0 because all heat waves are taken into account for each RCM. The ratio of the frequency of heat waves between CORDEX RCMs and TSMP (blue lines in Fig. 7b) increases toward intense heat waves ($\geq 5$ K). It shows a systematic behavior of TSMP to simulate less intense

heat waves on average in the focus domain compared to the CORDEX ensemble. The largest discrepancy is found between TSMP and CCLM driven by MPI-ESM-LR (blue solid line in Fig. 7b), up to a factor of 12 or even more, depending on the intensity considered. The REMO RCM, driven by different GCMs, shows the smallest differences to TSMP, while REMO2015 driven by IPSL-CM5A-LR simulates even less intense heat waves than TSMP. Overall, COSMO RCMs tend to simulate more intense heat waves than REMO RCMs.

The spatial distribution of the intense heat waves differs between GCM-RCMs, with their highest frequency in France and Scandinavia and the lowest in the Alps, on average in the CORDEX ensemble (Fig. 10, Fig. B3 in Appendix B). Note that the frequency is defined in relation to the total number of heat waves in each RCM. The mean frequency of intense heat waves in the focus domain ranges from the lowest value of 0.174 in REMO2015 driven by IPSL-CM5A-LR to the highest of 0.301

in CCLM driven by MPI-ESM-LR, i.e., 17.4-30.1 % of all simulated heat waves exceed the intensity of 5 K. Compared to CCLM driven by MPI-ESM-LR, TSMP simulates a lower frequency of intense heat waves in all PRUDENCE regions except Scandinavia. The largest discrepancies and the highest number of intense heat waves are in France, with TSMP simulating 24.6 % of all heat waves as intense and CCLM – 46.8 %, smallest differences are in Scandinavia, where TSMP simulates 19.3 % of all heat waves as intense and CCLM – 17.5 %. It is important to note that the regions with the highest number of heat

waves do not necessarily coincide with the regions with the highest number of intense heat waves (see Fig. 8 and Fig. 10). The origin of these differences should be further investigated and is beyond the scope of this analysis.

## 4 Summary and conclusions

In this study, we presented, in the context of dynamical downscaling of GCMs with RCMs experiment for climate change studies, a first-of-its-kind TSMP dataset forced by the CMIP5 MPI-ESM-LR GCM boundary conditions, where 3D groundwa-

ter hydrodynamics were explicitly represented. We studied the impact of groundwater coupling on the statistics of simulated heat events in regional historical climate simulations, with potential implications for climate change projections, by comparing TSMP results with the ensemble of EURO-CORDEX climate change scenario RCM control runs driven as well by CMIP5

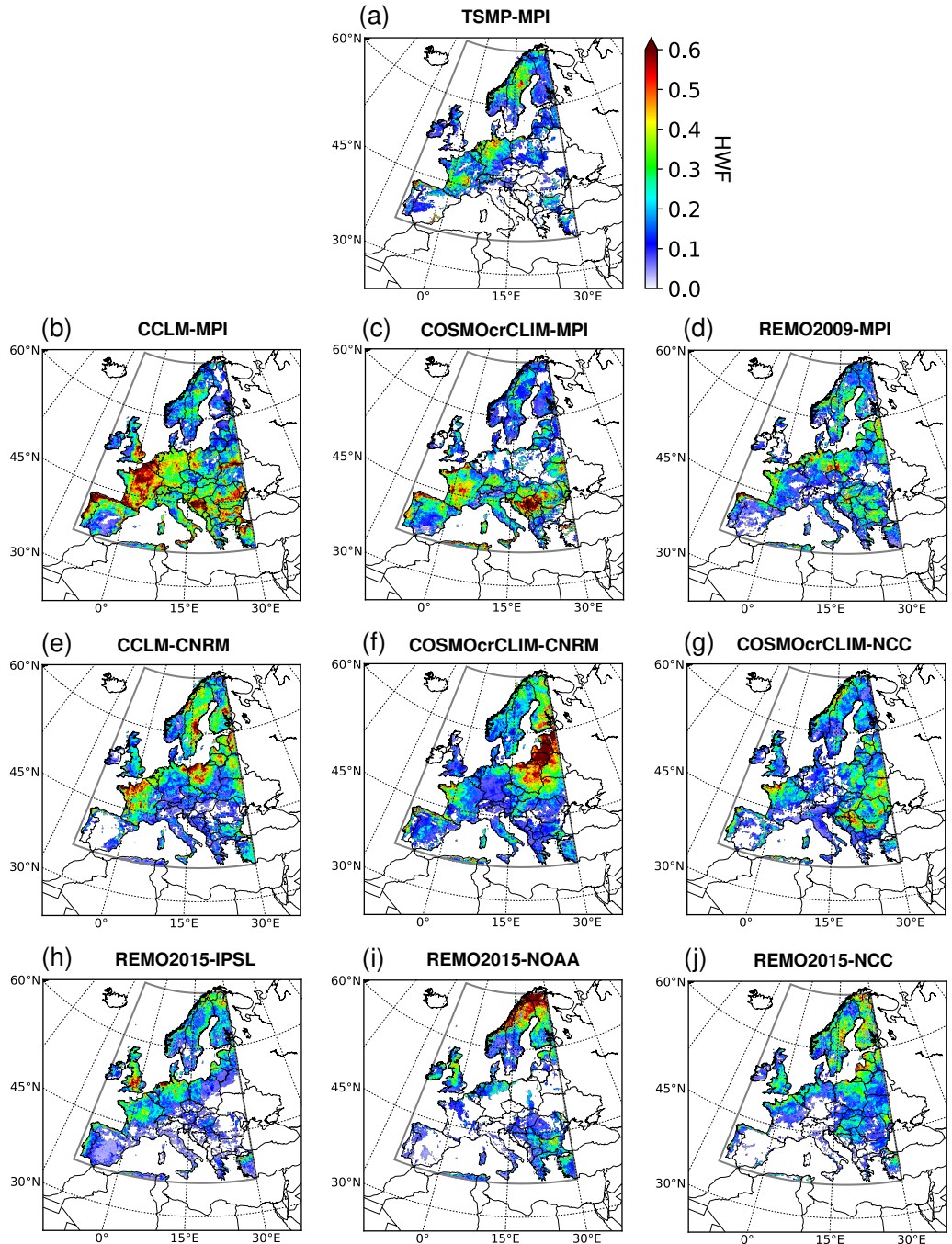

**Figure 10.** Frequency of heat waves (HWF) with an intensity above 5 K occurring between 1976 and 2005 with respect to the reference period 1961-1990, in the ensemble of EURO-CORDEX climate change scenario RCM control runs.

GCMs. In particular, we investigated the number of hot days and heat waves of different durations and intensities in Europe during the summer season between 1976 and 2005 relative to the reference period 1961-1990 in each RCM.

Our analysis shows that TSMP simulates heat events consistently with the EURO-CORDEX ensemble, although there are statistical differences and we relate these to groundwater coupling. TSMP simulates lower means as well as a lower interannual variability in the number of hot days on average in Europe, compared to the CORDEX ensemble. The decadal change in the number of hot days over Europe in TSMP is also lower compared to the CORDEX ensemble average. TSMP systematically simulates fewer heat waves and tends to simulate less intense heat waves compared to the CORDEX ensemble. The most

sensitive regions to groundwater coupling appear to be the Iberian Peninsula and the Mediterranean, while Scandinavia is the least sensitive.

    From a comparison of TSMP and CCLM driven by MPI-ESM-LR, with the largest differences in the COSMO lower boundary condition accounting for groundwater feedbacks in TSMP, we found that TSMP in the considered focus domain covering Europe simulates on average lower:

– mean number of hot days (TSMP – 10.95 days, CCLM – 11.80 days);

  – variability of the number of hot days (TSMP – 6.80 days, CCLM – 8.33 days);

  – decadal change in the number of hot days (TSMP – 1.53 days, CCLM – 1.86 days);

  – decadal number of heat waves (TSMP – 3.25, CCLM – 3.78);

  – contribution of heat waves to the number of hot days (TSMP – 22.38 %, CCLM – 24.96 %);

– frequency of intense heat waves (TSMP – 0.193, CCLM – 0.301).

    This study clearly indicates that a coupled regional climate system with a closed terrestrial water cycle, such as TSMP, systematically simulates a different climatology of heat event compared to uncoupled RCMs. The explicit representation of subsurface hydrodynamics and groundwater in RCMs may be key for the reduction of biases in the simulated duration and intensity of heat waves, particularly in Southern Europe. In the future, this work will be extended to investigate the evolution

of heat events under different climate change scenarios in TSMP compared to uncoupled RCMs and their control simulations.

# Appendix A: Number of hot days for different regions

| RCMs \ Regions | BI | IP | FR | ME | SC | AL | MD | EA | FD |
|---|---|---|---|---|---|---|---|---|---|
| TSMP-MPI | 10.95 | 10.36 | 11.50 | 11.76 | 10.00 | 12.20 | 11.40 | 11.13 | 10.95 |
| CCLM-MPI | 11.49 | 12.64 | 12.46 | 12.80 | 10.42 | 13.27 | 12.62 | 11.50 | 11.80 |
| COSMO-crCLIM-MPI | 9.94 | 12.54 | 11.28 | 11.18 | 9.93 | 12.91 | 12.03 | 10.82 | 11.10 |
| REMO2009-MPI | 9.45 | 12.75 | 10.82 | 10.90 | 10.24 | 11.56 | 11.59 | 11.28 | 11.14 |
| CCLM4-CNRM | 12.89 | 9.66 | 10.93 | 12.62 | 12.06 | 10.74 | 10.49 | 11.16 | 11.32 |
| COSMO-crCLIM-CNRM | 12.05 | 10.14 | 9.87 | 10.87 | 13.87 | 10.15 | 9.85 | 11.26 | 11.46 |
| COSMO-crCLIM-NCC | 12.78 | 11.66 | 11.21 | 10.96 | 12.84 | 10.76 | 9.02 | 9.91 | 11.10 |
| REMO2015-NCC | 12.61 | 12.89 | 12.06 | 12.86 | 13.31 | 13.27 | 11.27 | 12.95 | 12.72 |
| REMO2015-IPSL | 9.56 | 15.00 | 11.45 | 11.48 | 11.29 | 12.64 | 15.25 | 11.63 | 12.38 |
| REMO2015-NOAA | 9.96 | 15.33 | 10.54 | 8.98 | 12.21 | 11.56 | 13.87 | 10.05 | 11.77 |

**Figure A1.** Mean number of hot days, i.e., TG90p index, [days] for the summer season averaged between 1976 and 2005 with respect to the reference period 1961-1990 in the ensemble of EURO-CORDEX climate change scenario RCM control runs for the focus domain (FD) and the PRUDENCE regions: British Isles (BI), Iberian Peninsula (IP), France (FR), Mid-Europe (ME), Scandinavia (SC), Alps (AL), Mediterranean (MD), Eastern Europe (EA); see Fig. 4 for the spatial distribution.

| RCMs \ Regions | BI | IP | FR | ME | SC | AL | MD | EA | FD |
|---|---|---|---|---|---|---|---|---|---|
| TSMP-MPI | 7.68 | 6.17 | 6.92 | 7.43 | 6.26 | 7.50 | 6.93 | 6.93 | 6.80 |
| CCLM-MPI | 8.50 | 9.59 | 8.30 | 8.53 | 7.45 | 8.82 | 9.34 | 8.02 | 8.33 |
| COSMO-crCLIM-MPI | 6.75 | 8.01 | 6.56 | 6.90 | 6.56 | 8.56 | 9.00 | 7.42 | 7.38 |
| REMO2009-MPI | 6.73 | 8.94 | 7.36 | 6.81 | 6.48 | 7.39 | 9.37 | 7.62 | 7.62 |
| CCLM4-CNRM | 8.07 | 6.37 | 8.72 | 10.00 | 8.47 | 7.32 | 8.28 | 8.17 | 8.21 |
| COSMO-crCLIM-CNRM | 8.50 | 6.84 | 7.38 | 7.42 | 10.22 | 8.38 | 8.19 | 10.31 | 8.90 |
| COSMO-crCLIM-NCC | 9.24 | 6.89 | 7.91 | 7.41 | 10.06 | 8.73 | 8.79 | 7.59 | 8.37 |
| REMO2015-NCC | 9.94 | 8.05 | 7.21 | 8.57 | 12.21 | 8.48 | 8.86 | 8.77 | 9.42 |
| REMO2015-IPSL | 7.74 | 9.51 | 8.16 | 8.39 | 8.94 | 9.60 | 10.13 | 8.40 | 8.97 |
| REMO2015-NOAA | 6.88 | 8.71 | 6.18 | 5.87 | 10.24 | 8.05 | 10.49 | 8.11 | 8.69 |

**Figure A2.** Variability of the hot days number, i.e., TG90p index, [days] for the summer season from 1976 to 2005 with respect to the reference period 1961-1990 in the ensemble of EURO-CORDEX climate change scenario RCM control runs for the focus domain (FD) and the PRUDENCE regions: British Isles (BI), Iberian Peninsula (IP), France (FR), Mid-Europe (ME), Scandinavia (SC), Alps (AL), Mediterranean (MD), Eastern Europe (EA); see Fig. 4 for the spatial distribution.

| RCMs \ Regions | BI | IP | FR | ME | SC | AL | MD | EA | FD |
|---|---|---|---|---|---|---|---|---|---|
| TSMP-MPI | 0.62 | 2.26 | 2.90 | 1.52 | -0.98 | 3.33 | 3.74 | 1.84 | 1.53 |
| CCLM-MPI | 1.39 | 4.25 | 3.49 | 2.03 | -0.89 | 3.29 | 4.40 | 1.21 | 1.86 |
| COSMO-crCLIM-MPI | 0.79 | 3.66 | 2.87 | 1.98 | -0.62 | 4.02 | 2.86 | 1.47 | 1.68 |
| REMO2009-MPI | 1.80 | 3.69 | 3.29 | 2.67 | 0.15 | 2.50 | 2.65 | 1.29 | 1.88 |
| CCLM4-CNRM | 3.56 | 2.78 | 3.52 | 3.40 | 3.29 | 2.88 | 1.66 | 0.43 | 2.35 |
| COSMO-crCLIM-CNRM | 3.00 | 1.76 | 1.34 | 1.36 | 2.95 | 1.49 | 2.54 | 0.08 | 1.77 |
| COSMO-crCLIM-NCC | 4.50 | 1.58 | 3.56 | 2.43 | 4.09 | 1.32 | -1.24 | 0.52 | 1.87 |
| REMO2015-NCC | 3.92 | 2.64 | 2.23 | 2.51 | 3.08 | 3.17 | 1.97 | 2.85 | 2.71 |
| REMO2015-IPSL | 0.55 | 3.89 | 0.57 | 0.23 | 0.87 | 1.27 | 2.22 | -0.20 | 1.13 |
| REMO2015-NOAA | 1.11 | 5.18 | 1.44 | 0.13 | 4.27 | 1.27 | 3.49 | 1.16 | 2.68 |

**Figure A3.** Decadal change in the number of hot days, i.e., TG90p index, [days] in the summer season, based on data from 1976 to 2005 with respect to the reference period 1961-1990, in the ensemble of EURO-CORDEX climate change scenario RCM control runs for the focus domain (FD) and the PRUDENCE regions: British Isles (BI), Iberian Peninsula (IP), France (FR), Mid-Europe (ME), Scandinavia (SC), Alps (AL), Mediterranean (MD), Eastern Europe (EA); see Fig. 6 for the spatial distribution.

## 370 Appendix B: Heat waves characteristics for different regions

| RCMs \ Regions | BI | IP | FR | ME | SC | AL | MD | EA | FD |
|---|---|---|---|---|---|---|---|---|---|
| TSMP-MPI | 3.33 | 2.51 | 3.15 | 4.01 | 3.83 | 4.13 | 3.04 | 2.74 | 3.25 |
| CCLM-MPI | 3.85 | 4.88 | 4.20 | 3.84 | 3.59 | 4.50 | 4.48 | 2.86 | 3.78 |
| COSMO-crCLIM-MPI | 2.54 | 4.99 | 3.73 | 2.78 | 3.46 | 5.27 | 3.59 | 2.34 | 3.35 |
| REMO2009-MPI | 2.70 | 5.25 | 4.17 | 3.30 | 4.17 | 3.50 | 4.40 | 3.24 | 3.89 |
| CCLM4-CNRM | 5.25 | 2.53 | 3.57 | 5.31 | 4.69 | 5.09 | 4.43 | 4.15 | 4.31 |
| COSMO-crCLIM-CNRM | 4.84 | 3.18 | 3.69 | 4.09 | 5.65 | 4.03 | 4.18 | 5.27 | 4.63 |
| COSMO-crCLIM-NCC | 5.24 | 3.66 | 3.86 | 3.15 | 5.61 | 4.10 | 3.43 | 2.92 | 3.97 |
| REMO2015-NCC | 4.42 | 4.23 | 4.15 | 4.64 | 5.52 | 4.69 | 3.92 | 4.01 | 4.49 |
| REMO2015-IPSL | 3.21 | 7.25 | 4.49 | 4.23 | 4.44 | 5.59 | 6.77 | 4.58 | 5.09 |
| REMO2015-NOAA | 3.91 | 4.93 | 3.10 | 2.02 | 5.03 | 3.57 | 5.42 | 2.62 | 3.95 |

**Figure B1.** Decadal number of heat waves based on data from 1976 to 2005 with respect to the reference period 1961-1990, in the ensemble of EURO-CORDEX climate change scenario RCM control runs for the focus domain (FD) and the PRUDENCE regions: British Isles (BI), Iberian Peninsula (IP), France (FR), Mid-Europe (ME), Scandinavia (SC), Alps (AL), Mediterranean (MD), Eastern Europe (EA); see Fig. 8 for the spatial distribution.

| RCMs \ Regions | BI | IP | FR | ME | SC | AL | MD | EA | FD |
|---|---|---|---|---|---|---|---|---|---|
| TSMP-MPI | 25.36 | 17.47 | 19.62 | 25.61 | 29.87 | 24.40 | 19.33 | 17.79 | 22.38 |
| CCLM-MPI | 27.90 | 31.19 | 26.29 | 24.34 | 26.50 | 27.79 | 26.67 | 18.82 | 24.96 |
| COSMO-crCLIM-MPI | 18.78 | 31.96 | 25.50 | 18.49 | 27.59 | 33.62 | 24.46 | 16.27 | 23.46 |
| REMO2009-MPI | 21.81 | 33.04 | 28.35 | 22.39 | 32.63 | 23.50 | 30.18 | 21.92 | 27.22 |
| CCLM4-CNRM | 34.53 | 18.35 | 25.45 | 33.01 | 31.69 | 37.74 | 34.53 | 28.08 | 29.72 |
| COSMO-crCLIM-CNRM | 29.75 | 23.36 | 29.21 | 29.31 | 36.52 | 32.75 | 35.54 | 38.21 | 33.11 |
| COSMO-crCLIM-NCC | 34.70 | 23.94 | 30.71 | 23.12 | 39.04 | 30.05 | 33.07 | 23.70 | 29.61 |
| REMO2015-NCC | 30.16 | 25.39 | 32.05 | 29.06 | 38.37 | 29.72 | 27.54 | 26.69 | 30.20 |
| REMO2015-IPSL | 30.39 | 41.20 | 30.57 | 31.56 | 35.26 | 38.26 | 36.73 | 31.75 | 34.40 |
| REMO2015-NOAA | 30.77 | 25.20 | 21.41 | 16.51 | 37.52 | 24.58 | 32.54 | 19.36 | 26.97 |

**Figure B2.** Contribution of heat waves to the number of hot days [%] during 1976-2005 with respect to the reference period 1961-1990, in the ensemble of EURO-CORDEX climate change scenario RCM control runs for the focus domain (FD) and the PRUDENCE regions: British Isles (BI), Iberian Peninsula (IP), France (FR), Mid-Europe (ME), Scandinavia (SC), Alps (AL), Mediterranean (MD), Eastern Europe (EA); see Fig. 9 for the spatial distribution.

| RCMs \ Regions | BI | IP | FR | ME | SC | AL | MD | EA | FD |
|---|---|---|---|---|---|---|---|---|---|
| TSMP-MPI | 0.118 | 0.189 | 0.246 | 0.233 | 0.193 | 0.145 | 0.171 | 0.148 | 0.193 |
| CCLM-MPI | 0.246 | 0.305 | 0.468 | 0.395 | 0.175 | 0.326 | 0.318 | 0.319 | 0.301 |
| COSMO-crCLIM-MPI | 0.181 | 0.246 | 0.361 | 0.216 | 0.152 | 0.213 | 0.262 | 0.318 | 0.241 |
| REMO2009-MPI | 0.173 | 0.156 | 0.207 | 0.230 | 0.211 | 0.125 | 0.180 | 0.218 | 0.202 |
| CCLM4-CNRM | 0.185 | 0.199 | 0.356 | 0.244 | 0.290 | 0.121 | 0.125 | 0.218 | 0.233 |
| COSMO-crCLIM-CNRM | 0.111 | 0.187 | 0.236 | 0.157 | 0.309 | 0.106 | 0.143 | 0.293 | 0.232 |
| COSMO-crCLIM-NCC | 0.160 | 0.188 | 0.234 | 0.163 | 0.193 | 0.105 | 0.165 | 0.268 | 0.204 |
| REMO2015-NCC | 0.181 | 0.173 | 0.163 | 0.166 | 0.267 | 0.082 | 0.123 | 0.184 | 0.205 |
| REMO2015-IPSL | 0.287 | 0.112 | 0.219 | 0.214 | 0.203 | 0.112 | 0.084 | 0.108 | 0.174 |
| REMO2015-NOAA | 0.213 | 0.094 | 0.120 | 0.197 | 0.294 | 0.101 | 0.144 | 0.141 | 0.213 |

**Figure B3.** Frequency of heat waves with intensity exceeding 5 K based on data from 1976 to 2005 with respect to the reference period 1961-1990, in the ensemble of EURO-CORDEX climate change scenario RCM control runs for the focus domain (FD) and the PRUDENCE regions: British Isles (BI), Iberian Peninsula (IP), France (FR), Mid-Europe (ME), Scandinavia (SC), Alps (AL), Mediterranean (MD), Eastern Europe (EA); see Fig. 10 for the spatial distribution.

*Code and data availability.* The TSMP v1.2.2 used in this work is available through https://github.com/HPSCTerrSys/TSMP GIT repository.

The dataset from the MPI-ESM-LR r1i1p1 GCM driven TSMP can be accessed at https://datapub.fz-juelich.de/slts/regional_climate_tsmp_ hi-cam/ as open access research data.

*Author contributions.* The study was designed by S.K. with contributions by K.G., L.P-S., and N.W.. L.P.-S. performed the model simulations and data processing, N.W. provided technical and programming support, C.H. provided setups, configuration and workflow support. The analysis was conducted by L.P.-S. with further inputs from S.K. and K.G.. L.P.-S. wrote the manuscript. All co-authors contributed to the

interpretation of the results, active discussions, and revisions of the paper. The work was done under the supervision of S.K..

*Competing interests.* The authors declare that they have no conflict of interest.

*Acknowledgements.* This work was funded by the Helmholtz Association of German Research Centres (HGF) under the HI-CAM project (Helmholtz Initiative Climate Adaptation and Mitigation) and by the German Ministry of Education and Research (Bundesministerium für Bildung und Forschung, BMBF) under the ClimXtreme project. We thank the EURO-CORDEX climate modelling groups for producing

and making available their model output. We are grateful to the Max-Planck Institute for performing MPI-ESM-LR r1i1p1 GCM experiment and the German Climate Computing Centre (DKRZ) for providing the MPI-ESM-LR dataset. The authors gratefully acknowledge the Earth System Modelling Project (ESM) for funding this work by providing computing time on the ESM partition of the supercomputer JUWELS at the Jülich Supercomputing Centre (JSC) under the ESM project ID JIBG35. In addition, we thank the Centre for High-Performance Scientific Computing in Terrestrial Systems (Geoverbund ABC/J, http://www.hpsc-terrsys.de) and the JSC for the computational support.

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
