# Peer review of "Impact of 3D groundwater dynamics on heat events in historical regional climate simulations over Europe"

_Earth System Dynamics, 2022_

## Author Comment (AC1)

**Reply to Referee #1**
**Groundwater in terrestrial systems modelling: a new climatology of extreme heat events in Europe**

**We thank the referee for the review and for the helpful and detailed comments. We provide a point-by-point reply below, where the reviewer comments are repeated in black. The replies to the reviewer's comments are in blue colour. The revised text is given in italics and in quotation marks.**

**General remarks**

The study aims to improve the understanding of how groundwater affects regional temperature anomalies by comparing the simulation results from TSMP and those from the other RCMs. Although the topic and method are of interest to the readership of Earth System Dynamics, several issues need to be addressed.

- First, the manuscript is poorly written because of many confusing and inaccurate sentences.

    Thank you for the comment. We are carefully revising the manuscript for content, structure, language and grammar.

- Second, the results are not new since there already have been comparations in heat event analyses between TSMP and the other RCMs.

    Thank you for this comment. We agree that the main message of the paper was not clearly presented. Therefore we will revise the paper following the rational below.

    In the paper, we present a new dataset from the coupled Terrestrial Systems Modelling Platform (TSMP), driven by MPI-ESM-LR GCM, in the context of EURO-CORDEX GCM-RCM long-term climate modelling, and, in particular, the climate change scenario control runs (until 2006). While the evaluation of TSMP driven by the ERA-Interim reanalysis has already been performed in the work of Furusho-Percot et al. (2019, 2022), long-term historic climate simulations of TSMP forced by GCM boundary information have not been previously presented. This is the first downscaled regional historic climate simulation from groundwater across the land surface to the top of atmosphere, which has been put into context of the large EURO-CORDEX ensemble and analyzed for extreme heat events in this study.

    We see a high potential for a dataset like TSMP driven by MPI-ESM-LR GCM, with explicit representation of groundwater dynamics. The characteristics of summer heat events from TSMP compared to the CORDEX ensemble members with oversimplified or

neglected groundwater dynamics, discussed in this paper, not only adds important information to the existing ensemble, but also of existing uncertainties between the CORDEX ensemble due to explicit groundwater inclusion, which is essential for drawing conclusions about the uncertainties to be expected in projection analyses. Also in the light of current groundwater drought in Central and Southern Europe, the results are of importance in the assessment of future temperature extremes.

We have added this paragraph to the introduction of the revised paper:
*"In this paper, we present a unique dataset from TSMP forced by the Max Planck Institute Earth System Model with Low Resolution MPI-ESM-LR (Giorgetta et al., 2013) historical boundary information in the context of EURO-CORDEX GCM-RCM long-term climate modelling, and, in particular, the climate change scenario control runs. We interrogate the statistics of the heat event characteristics (frequency, duration, intensity) of 1976-2005 with respect to the reference period 1961-1990 by comparing TSMP results with the EURO-CORDEX multi-model RCM ensemble driven by CMIP5 (Taylor et al., 2012) GCM control simulations, to understand the influence of 3D groundwater dynamics on simulated heat extremes for regional historical climate simulations and potential consequences for climate change projections. While the 1996-2018 TSMP evaluation runs nested within ERA-Interim reanalysis were examined for heat wave statistics (Furusho-Percot et al., 2019, 2022), long-term historical climate simulations of TSMP forced by GCM boundary information have not been previously presented. Thus, this is the first downscaled regional historical climate simulations from groundwater across the land surface to the top of the atmosphere placed in the context of the EURO-CORDEX ensemble and analyzed for extreme summer heat events."*

- Meanwhile, the key results showing characteristics of heat waves/events are not well organized for the readers to follow.
  Thank you for your comment. We agree that this is an important point and the reorganisation of the Results section will make the manuscript clearer to the reader. We will review the Results section according to your suggestions in the revised version of the paper.

- Third, comparisons between the simulation results of TSMP and the ensemble of RCMs (Figure 5) are not convincing to conclude the influence of groundwater, since the different conceptualizations of physical processes among the models can also result in differences in temperature anomaly, let alone different forcing data (Table 1).
  We agree that the differences between the various RCMs and driving GCMs need to be honored in the analyses and discussion when making the comparison. At the beginning of Section 3.2, we discussed these limitations already:
  *"Due to connections of various factors other than groundwater coupling in the multi-model CORDEX ensemble (e.g., various model setups, conceptual and structural model uncertainties, different physical parameterizations, internal variability, representation of subsurface-land-atmosphere interactions, lower and lateral atmospheric GCM boundary conditions), it is challenging to reveal the exact cause and effect relationship of the explicit groundwater representation for simulated hot days and the associated heat events*

*characteristics in RCMs. Moreover, the ensemble of EURO-CORDEX climate change scenario RCM control runs is not intended for direct comparison between individual models, as it includes different RCMs in combination with different driving GCMs. However, as has been shown in previous studies, the consideration of an extended period, e.g., 30-years, allows to draw statistically conclusions."*

To improve the manuscript further, we will (1) expand the discussion on the selection of RCMs-GCMs in the CORDEX ensemble, (2) clearly formulate the main objectives of the paper, stating that we want to provide an overview of whether new GCM-RCM TSMP-MPI dataset is consistent with the CORDEX ensemble and make a statement on the role of groundwater inclusion in RCMs for long-term climate simulations on the example of heat waves statistics, as it is essential to draw conclusions about the uncertainties expected in projection analyses (please see our response to point #2 of "General remarks").

We would like to point out that Keune et al. (2016) with dedicated TSMP simulations (with and without 3D groundwater flow) clearly demonstrated the impact of groundwater on the land surface water and energy balance including temperature. In this study, the model was used in the same version (including groundwater) and with improved geology and topographic slopes. Repeating the dedicated simulations with/without groundwater at the climate time scale is computationally not feasible. Our rational and its limitations will be discussed in detail in the revised manuscript.

- Fourth, TSMP has the advantage of fully characterizing groundwater-soil moisture-temperature interactions. Still, the manuscript fails to analyze more variables of the model outputs rather than temperature anomalies. It fails to investigate why and how groundwater can affect the characteristics of extreme heat events.

  Thank you for this comment. As outlined in the answer above, the main objectives of this paper are not to demonstrate the impact of groundwater on temperatures, which has already been done in previous studies (e.g., Barlage et al., 2015; Keune et al., 2016). Instead we want to provide an overview of whether new GCM-RCM TSMP-MPI dataset is consistent with the CORDEX ensemble and arrive at a statement on the role of groundwater in RCMs for long-term climate simulations on the example of heat waves statistics.

  We agree with the referee that the link between groundwater and its impact on temperatures was not addressed well in the last version of the manuscript and the discussion of the previous studies on TSMP was missing. Thus, taking all this into account, we (1) have extended the introduction of the paper to include an overview of previous studies on TSMP (see below) and (2) clearly stated the main objectives in the revised version of the paper (please see our response to point #2 of "General remarks").

  *"The role of soil moisture in modelling extreme heat events is crucial (e.g., Seneviratne et al., 2006, 2010; Fischer et al., 2007), but due to the complexity of the feedbacks involved and related high computational cost, the explicit representation of hydrological processes is oversimplified or neglected in most RCMs. Commonly applied hydrology schemes are based on 1D-parameterizations in the vertical direction with gravity free drainage approach as the boundary condition at the bottom and runoff generation at the*

*land surface; in such a parametrisation there is no lateral subsurface flow and only the 1D-Richards' equation is solved (e.g., Niu et al., 2007; Campoy et al., 2013). RCMs with simplified representation of hydrological processes are unable to reliably reproduce land energy flux partitioning and, consequently, near-surface air temperatures, leading to warm biases (Vautard et al., 2013a; Barlage et al., 2021; Furusho-Percot et al., 2022). Hydrological parameters tuning (e.g., Teuling et al., 2009; Bellprat et al., 2016) or developing new parameterizations of groundwater dynamics (e.g., Liang et al., 2003; Yeh and Eltahir, 2005; Schlemmer et al., 2018) have been shown to improve model results. A physically consistent description of hydrological processes in RCMs can be achieved by an explicit representation of 3D subsurface and groundwater hydrodynamic together with overland flow, and accounting for a complete feedback loop over the terrestrial system (e.g., Maxwell et al., 2007), i.e., water and energy cycles from groundwater across the land surface to the top of the atmosphere, as in the regional Terrestrial Systems Modelling Platform (TSMP) (Shrestha et al., 2014; Gasper et al., 2014).*

*TSMP is a scale-consistent, highly modular, fully integrated soil-vegetation-atmosphere coupled regional climate model. TSMP comprises the hydrological model ParFlow v.3.2 (e.g., Kollet and Maxwell, 2008; Maxwell, 2013), the Community Land Model (CLM) v.3.5, and the atmospheric model Consortium for Small Scale Modelling (COSMO) v.5.01 (e.g., Baldauf et al., 2011), which are coupled externally via the Ocean Atmosphere Sea Ice Soil (OASIS, version 3.0) Model Coupling Toolkit (MCT) (e.g., Valcke, 2013) to exchange fluxes between independent component models of TSMP. Keune et al. (2016) demonstrated the link between groundwater and near-surface temperature in an analysis of the August 2003 European heat wave from the TSMP simulations nested within ERA-Interim (Dee et al., 2011) and set up over the the European domain of the COordinated Regional Downscaling EXperiment (EURO-CORDEX) (Gutowski et al., 2016; Jacob et al., 2020), with two different groundwater configurations: (i) simplified 1D free drainage approach and (ii) 3D physics-based variably saturated groundwater dynamics. The study clearly showed an impact of groundwater dynamics on the land surface water and energy balance: latent heat fluxes were higher and maximum temperatures were lower, especially in areas with shallow water table depth, in the 3D configuration compared to the simplified 1D free drainage approach. Keune et al. (2016) suggest that the 3D groundwater dynamics in TSMP alleviate the evolution of heat extremes due to weaker land-atmosphere feedbacks compared to the simplified 1D free drainage approach, at least during the investigated European heat wave of summer 2003. The ability of groundwater to moderate warm summer biases and decrease maximum air temperatures during a single seasonal heat wave in RCM simulations was also discussed in Barlage et al. (2015, 2021) and Mu et al. (2022).*

*As an explanation, the 3D groundwater dynamics in TSMP leads to shallower groundwater levels compared to 1D approach, causing wetter soils, and a reduction in the Bowen ratio (sensible heat flux to latent heat flux) due to an increase in surface latent heat flux and a decrease in surface sensible heat flux, i.e., an increase in evapotranspiration (Maxwell and Condon, 2016). On the one hand, such an increase in a latent heat flux*

*causes moistening of the lower atmosphere and increases downward longwave radiation due to the greenhouse effect of water vapor, on the other hand, it cools the surface and reduces outgoing surface longwave radiation (Pal and Eltahir, 2001). In addition, increased evapotranspiration may cause moist convection or rainfall, which further affects soil moisture (Eltahir, 1998; Yang et al., 2018). In its turn, the simplified representation of groundwater dynamics with the 1D free drainage approach leads to the opposite effect, namely an overestimation of the land surface-atmosphere coupling via shallow soil moisture and strengthening of the feedback mechanisms, i.e., deeper groundwater levels cause drier soils, an increase in the Bowen ratio by reducing latent and increasing sensible heat fluxes, a decrease in cloud cover and enhance of incoming shortwave radiation, and, as a result, higher near-surface temperatures, which in turn further enhances latent heat flux and reduces soil moisture (Vogel et al., 2018).*

*Further studies were carried out to understand whether the observed differences in simulated near-surface temperature due to differences in groundwater configuration (3D physics-based in TSMP and simplified in RCM ensemble) persist over longer time periods, and how this manifests itself for heat waves in the EURO-CORDEX realm. Furusho-Percot et al. (2019) showed that TSMP evaluation run (1996–2018) forced by the ERA-Interim reanalysis is able to capture climate system dynamics and the succession of warm and cold seasons on the regional scale for the PRUDENCE regions of Europe (Christensen and Christensen, 2007) consistently with E-OBS observations (Cornes et al., 2018). Another study by Furusho-Percot et al. (2022) demonstrated that TSMP multiannual simulations exhibit lower absolute deviations of summer heat wave indices from the E-OBS observational dataset, compared to ERA-Interim-driven RCM evaluation simulations of the EURO-CORDEX experiment (Jacob et al., 2020), which tend to simulate too persistent heat waves (Vautard et al., 2013a). This particular behaviour of TSMP is attributed to the improved hydrology due to the explicit representation of 3D groundwater dynamics, namely the improved capacity to sustain soil moisture translates into more reliable latent heat flux and evapotranspiration, that, in turn, leads to a decrease in the heat wave amplitude, extent and the number of days with anomalously high near-surface temperatures, unlike in the CORDEX RCM ensemble with simplified groundwater representation. An important question remains: how will these findings be reflected in long-term regional climate simulations?*

*In this paper, we present a unique dataset from TSMP forced by the Max Planck Institute Earth System Model with Low Resolution MPI-ESM-LR (Giorgetta et al., 2013) historical boundary information in the context of EURO-CORDEX GCM-RCM long-term climate modelling, and, in particular, the climate change scenario control runs. We interrogate the statistics of the heat event characteristics (frequency, duration, intensity) of 1976-2005 with respect to the reference period 1961-1990 by comparing TSMP results with the EURO-CORDEX multi-model RCM ensemble driven by CMIP5 (Taylor et al., 2012) GCM control simulations, to understand the influence of 3D groundwater dynamics on simulated heat extremes for regional historical climate simulations and potential consequences for climate change projections. While the 1996-2018 TSMP evaluation runs*

*nested within ERA-Interim reanalysis were examined for heat wave statistics (Furusho-Percot et al., 2019, 2022), long-term historical climate simulations of TSMP forced by GCM boundary information have not been previously presented. Thus, this is the first downscaled regional historical climate simulations from groundwater across the land surface to the top of the atmosphere placed in the context of the EURO-CORDEX ensemble and analyzed for extreme summer heat events."*

**Specific Comments**

**Title of the manuscript**

- Title: Is that suitable to say "climatology of extreme heat events"? since this work is talking about the climatological characteristics/mechanisms of extreme heat events.
  Thank you for the remark. We revised the title as follows *"The influence of 3D groundwater dynamics on heat events in a regional historic climate simulations"*.

**Abstract**

1. Line 1: This sentence is not concise. Particularly, it seems unnecessary to talk about the requirements of high-resolution climate data, which is not the point of this study's focus. The potential background might be why groundwater processes can affect extreme heat events for the second sentence.
   Thank you for the suggestion. We rephrased the sentence in the revised version of the paper.
   *"The representation of groundwater processes is simplified in most regional climate models (RCMs), potentially leading to biases in simulated extreme heat events in Europe."*

2. Line 4: What do you mean by a unique dataset? How unique?
   Thank you for the specific questions. TSMP is a scale-consistent, fully integrated soil-vegetation-atmosphere modelling system. The uniqueness of TSMP lies in the explicit representation of three-dimensional subsurface and groundwater hydrodynamics together with overland flow. TSMP closes the water and energy cycle from the bedrock to the top of the atmosphere. This makes TSMP and the data set unique in our opinion.

   We have added this information to the abstract of the revised version of the paper:
   *"Here, we introduce a unique dataset from the regional Terrestrial Systems Modeling Platform (TSMP) forced by the Max Planck Institute Earth System Model with Low Resolution (MPI-ESM-LR) historical boundary information in the context of dynamical downscaling of global climate models (GCMs) with RCMs for climate change studies. TSMP explicitly represents the three-dimensional subsurface and groundwater hydrodynamics together with overland flow, closing the water and energy cycle from the bedrock to the top of the atmosphere."*

3. Line 5: Not all readers know what EURO-CORDEX is.
Thank you for the remark. We added the explanation to the revised version of the paper:
*"... compared to an ensemble of GCM-RCM simulations with oversimplified or neglected groundwater dynamics from the Coordinated Regional Climate Downscaling Experiment for European domain (EURO-CORDEX)".*

4. Line 6: It should be "period of 1976-2005"
Thank you for the remark. We have removed the sentence you are referring to from the abstract.

5. Line 8: What is the respective evaporative regime? And in the following sentences, regional differences are not mentioned.
Thank you for the specific correction. The information about the evaporative regimes was added in the revised version.
*"The impact of groundwater coupling on the frequency of hot summer days depends on the considered time period and region, associated with different evaporative regimes: energy-limited (predominant in Scandinavia and Northeastern Europe) versus moisture-limited (Southern and Southeastern Europe)."*

6. Line 9: Are "the other RCMs considered" just the CORDEX ensemble?
Yes, in addition to TSMP, we considered the CORDEX ensemble. To avoid confusion, we rephrased the sentence as follows:
*"An increasing trend of the frequency of hot summer days averaged across Europe is the lowest in TSMP compared to the CORDEX ensemble."*

7. Line 13: Not rigorous for using "slightly more"
Thank you for the remark. We rephrased the sentence, please see the revised version.
*"In particular, extended heat events with a duration exceeding 6 days (i.e. heat waves) occur on average in Europe about 1.5-8 times less often in TSMP, while single-day heat events happen more often in TSMP compared to the CORDEX ensemble."*

8. Line 13: Although it has been explained that a heat event with a duration exceeding 6 days indicates a heat wave, what are high-intensity heat waves?
Thank you for this question. We do not give a definition of high intensity heat waves, we only emphasise the fact that in the EURO-CORDEX ensemble there is a large spread of heat wave frequencies, and this spread increases towards heat waves of high intensity (Fig. 6b). At the same time, TSMP exhibits behaviour with fewer heat waves of high intensity compared to the CORDEX ensemble. For consistency, we have removed the sentence you are referring to from the abstract.

9. Line 15: First, using simplified might be better than using "explicit"? Second, it seems hard to conclude "rarer and weaker heat waves" since the frequency of heat waves and high-intensity heat waves is higher for the CORDEX ensemble, where only single-day heat events happen less often. Third, as mentioned above, are the RCMs in this study mean the CORDEX ensemble?

Thank you for these remarks and suggestions.
We do not agree with the referee that the word "explicit" in the text could be replaced by "simplified", as this would completely change the meaning of the sentence.

We agree with the referee that the phrasing "rarer" is incorrect, we have rephrased the sentence in the revised version of the paper.
*"Therefore, an explicit groundwater representation in CORDEX RCMs may lead to shorter and weaker heat waves in Europe."*

Regarding the third question, please see the answer to point #6.

10. Line 15: what do you mean about "also in climate projections"?
    Thank you for the remark. We have removed the sentence you are referring to from the abstract.

**Introduction**

1. Lines 44-49: These are not informative or conclusive enough to be an independent paragraph.
   Thank you for the remark. We rewrote this paragraph in the revised version.
   *"In the context of climate impact assessments, dynamical downscaling of global climate models (GCMs) with regional climate models (RCMs) is widely used to generate regional climate change scenarios (e.g., Vautard et al., 2013b; Mearns et al., 2015). Although dynamical downscaling technique with RCMs have been shown to provide added value to driving GCMs by better capturing small-scale processes and features (Giorgi and Gutowski, 2015; Torma et al., 2015; Prein et al., 2016; Iles et al., 2020; Rummukainen, 2016), model biases (offset during the historical period against observations) and uncertainties in climate projections still remain (Hawkins and Sutton, 2009; Lhotka et al., 2018; Sørland et al., 2018; Fernandez-Granja et al., 2021). In fact, many RCMs tend to overestimate the frequency, duration, and intensity of extreme heat events (Vautard et al., 2013a; Plavcová and Kyselý, 2016; Lhotka et al., 2018; Furusho-Percot et al., 2022)."*

   Information about the EURO-CORDEX domain and PRUDENCE regions, which was previously in the same paragraph, will be given in other paragraphs throughout the text.

2. Overall, this section is relatively clear about the importance of soil moisture in groundwater-soil moisture-temperature coupling, which affects temperature anomaly, and the advantage of groundwater parameterizations in TSMP. However, it is not firmly linked to the manuscript's aim to study the statistics of the heat event characteristics (frequency, duration, intensity).
   Thank you for this comment. We have rewritten the introduction as suggested, to more clearly demonstrate to the reader the main objectives of this paper. Please see the answer (above) to the point #1 of the introduction section, as well as the last point of the "General remarks".

**Methods**

1. The introduction of TSMP could be simplified, while the setup of TSMP could be clearer for reproducibility.
   Thank you for the comment. We are working on modifying sub-sections 2.1 and 2.2 of the Methods as suggested.

2. Line 105 and 108: It is unclear in which period the model was warmed up.
   Thank you for this remark. It is stated in the text already that
   *"TSMP climate change scenario control simulations were conducted from December 1949 to the end of 2005. Land surface, subsurface hydrology, and energy states were initialized with the moisture conditions of 1st of December 2011 from a spun-up evaluation run driven by ERA-Interim reanalysis (Furusho-Percot et al., 2019)."*

   In addition, we have added the following clarifications to the text:
   *"As a result, we discarded the first decade of TSMP climate change scenario control simulations in the analysis due to hydrodynamic spin-up. Thus, in the analyses the summer months of 1961-2005 were used."*

3. Lines 112-115: Is the model calibrated by well observations of the water table or observation-based surface temperature datasets? And how accurate are the simulation results?
   Thank you for the questions. The model was not calibrated. The evaluation run of TSMP was performed with atmospheric forcing derived from the ERA-Interim reanalysis. The temperature and precipitation fields simulated by TSMP were validated by comparison with the European Climate Assessment and Dataset (E-OBS v19). The Pearson's correlation values between the simulated and observed data series showed a good agreement in most European regions, with scores ranging from 0.73 to 0.94 for mean temperature anomalies and from 0.62 to 0.88 for precipitation anomalies. Please see Furusho-Percot et al. (2019) for more details.

   In the recent publication, Ma et al. (2022) used the TSMP water table simulation results in a machine learning approach and compared to in-situ water table observation anomalies collected over Europe. The results showed good agreement considering that this model has not been calibrated as aforementioned.

   For clarity, we will add this information to the TSMP description in the Methods section of the revised version of the manuscript.

**References**

[revised manuscript text omitted]

---

## Author Comment (AC2)

**Reply to Referee #2**
**Groundwater in terrestrial systems modelling: a new climatology of extreme heat events in Europe**

**We thank the referee for the review and for the helpful comments and suggestions. We provide a point-by-point reply below, where the reviewer comments are repeated in black. The replies to the reviewer's comments are in blue. The revised text is given in italics and in quotation marks.**

**General remarks**

Poshyvailo-Strube and colleagues investigated how the inclusion of groundwater modeling would affect heat wave characteristics in Europe. This is an important topic as groundwater plays a critical role in land-atmosphere interactions, but its modeling has been oversimplified or neglected in climate models. However, before I recommend publication, I have two major concerns that need to be addressed by the authors.

1. The title seems to suggest that this paper will be providing and evaluating a new climatology of extreme heat events in Europe.
   Thank you for this comment. The current title does not fully describe the main message we wanted to convey to the reader. Therefore, we decided to revise the title of the paper as follows *"The influence of 3D groundwater dynamics on heat events in a regional historic climate simulations"*.

2. I would expect the authors to compare their new climatology to observation-informed heat wave characteristics. However, I did not find any comparison with observations. The paper itself seems to discuss how groundwater modeling would affect heat waves just by comparing their model to other nogroundwater models. This seems not new to me as the authors have stated in line 60. An opportunity to improve probably is to add an observational perspective.
   In the paper, we present a new dataset from the coupled Terrestrial Systems Modelling Platform (TSMP), driven by MPI-ESM-LR GCM historical boundary information. The simulation was performed, in the context of GCM-RCM EURO-CORDEX long-term climate modelling, and, in particular, the climate change scenario control runs (until 2006). Thus, the model is informed at the boundaries with data from a GCM, and the simulation results can not be evaluated directly with observations. However, the evaluation of TSMP driven by the ERA-Interim reanalysis against observations has already been performed in the work of Furusho-Percot et al. (2019, 2022), which showed good agreement. This also holds for the large scale water budgets and anomalies (Hartick et al., 2021; Ma et al., 2022).

Long-term historic climate simulations of TSMP forced by GCM boundary information have not been previously presented. This is the first downscaled regional historic climate simulation from groundwater across the land surface to the top of the atmosphere, which has been put into context of the large EURO-CORDEX ensemble and analyzed for extreme heat events in this study.

We see a high potential for a dataset like TSMP driven by MPI-ESM-LR GCM, with explicit representation of groundwater dynamics. The characteristics of summer heat events from TSMP compared to the CORDEX ensemble members with oversimplified or neglected groundwater dynamics, discussed in this paper, not only adds important information to the existing ensemble, but also of existing uncertainties between the CORDEX ensemble due to explicit groundwater inclusion, which is is essential for drawing conclusions about the uncertainties to be expected in projection analyses. Also in the light of current groundwater drought in Central and Southern Europe, the results are of importance in the assessment of future temperature extremes.

In our opinion, this line of arguments was not clearly presented in the paper. Therefore, we have revised the introduction of the paper, following the rational above, and added a clear explanation of the objectives of the paragraph:
*"...An important question remains: how will these findings be reflected in long-term regional climate simulations?*
*In this paper, we present a unique dataset from TSMP forced by the Max Planck Institute Earth System Model with Low Resolution MPI-ESM-LR (Giorgetta et al., 2013) historical boundary information in the context of EURO-CORDEX GCM-RCM long-term climate modelling, and, in particular, the climate change scenario control runs. We interrogate the statistics of the heat event characteristics (frequency, duration, intensity) of 1976-2005 with respect to the reference period 1961-1990 by comparing TSMP results with the EURO-CORDEX multi-model RCM ensemble driven by CMIP5 (Taylor et al., 2012) GCM control simulations, to understand the influence of 3D groundwater dynamics on simulated heat extremes for regional historical climate simulations and potential consequences for climate change projections. While the 1996-2018 TSMP evaluation runs nested within ERA-Interim reanalysis were examined for heat wave statistics (Furusho-Percot et al., 2019, 2022), long-term historical climate simulations of TSMP forced by GCM boundary information have not been previously presented. Thus, this is the first downscaled regional historical climate simulations from groundwater across the land surface to the top of the atmosphere placed in the context of the EURO-CORDEX ensemble and analyzed for extreme summer heat events."*

3. Are the heat wave characteristics modeled by TSMP better agree with temperature observations than other models and by how much?
Thank you for the questions. In the recent work by Furusho-Percot et al. (2022), it was shown that multiannual heat wave statistics from TSMP simulations forced by the ERA-Interim reanalysis are consistent with E-OBS observations and the ERA5 reanalysis. Moreover, TSMP heat wave metrics (intensity, extent, number of heat wave days) have consistently shown lower mean absolute deviations from observations compared to

RCMs with simplified groundwater representation, exhibiting a warm bias. It is explained by the explicit representation of 3D groundwater dynamics in TSMP.

For clarity, we have added this text in the revised version:
*"... Furusho-Percot et al. (2019) showed that TSMP evaluation run (1996–2018) forced by the ERA-Interim reanalysis is able to capture climate system dynamics and the succession of warm and cold seasons on the regional scale for the PRUDENCE regions of Europe (Christensen and Christensen, 2007) consistently with E-OBS observations (Cornes et al., 2018). Another study by Furusho-Percot et al. (2022) demonstrated that TSMP multiannual simulations exhibit lower absolute deviations of summer heat wave indices from the E-OBS observational dataset, compared to ERA-Interim-driven RCM evaluation simulations of the EURO-CORDEX experiment (Jacob et al., 2020), which tend to simulate too persistent heat waves (Vautard et al., 2013). This particular behaviour of TSMP is attributed to the improved hydrology due to the explicit representation of 3D groundwater dynamics, namely the improved capacity to sustain soil moisture translates into more reliable latent heat flux and evapotranspiration, that, in turn, leads to a decrease in the heat wave amplitude, extent and the number of days with anomalously high near-surface temperatures, unlike in the CORDEX RCM ensemble with simplified groundwater representation."*

4. The current paper lacks an investigation of which process the groundwater had the influence to change the temperature anomalies. The intuitive processes are soil moisture and evapotranspiration...
Thank you for this comment. We agree with the referee that the link between groundwater and its impact on temperatures was not described well in the last version of the manuscript. In particular, the discussion of the previous studies on TSMP was missing. Note that the main objectives of this paper are not to demonstrate the impact of groundwater on temperatures, which has already been done in previous studies (e.g., Barlage et al., 2015; Keune et al., 2016). Instead we want to provide an overview of whether new GCM-RCM TSMP-MPI dataset is consistent with the CORDEX ensemble and arrive at a statement on the role of groundwater in RCMs for long-term climate simulations on the example of heat waves statistics.

Taking all this into account, we (1) have extended the introduction of the paper in its revised version to include an overview of previous studies on TSMP (see below), (2) clearly stated the main objectives of the paper in its revised version (please see our response to point #2 of "General remarks").

*"The role of soil moisture in modelling extreme heat events is crucial (e.g., Seneviratne et al., 2006, 2010; Fischer et al., 2007), but due to the complexity of the feedbacks involved and related high computational cost, the explicit representation of hydrological processes is oversimplified or neglected in most RCMs. Commonly applied hydrology schemes are based on 1D-parameterizations in the vertical direction with gravity free drainage approach as the boundary condition at the bottom and runoff generation at the land surface; in such a parametrisation there is no lateral subsurface flow and only the*

*1D-Richards' equation is solved (e.g., Niu et al., 2007; Campoy et al., 2013). RCMs with simplified representation of hydrological processes are unable to reliably reproduce land energy flux partitioning and, consequently, near-surface air temperatures, leading to warm biases (Vautard et al., 2013; Barlage et al., 2021; Furusho-Percot et al., 2022). Hydrological parameters tuning (e.g., Teuling et al., 2009; Bellprat et al., 2016) or developing new parameterizations of groundwater dynamics (e.g., Liang et al., 2003; Yeh and Eltahir, 2005; Schlemmer et al., 2018) have been shown to improve model results. A physically consistent description of hydrological processes in RCMs can be achieved by an explicit representation of 3D subsurface and groundwater hydrodynamic together with overland flow, and accounting for a complete feedback loop over the terrestrial system (e.g., Maxwell et al., 2007), i.e., water and energy cycles from groundwater across the land surface to the top of the atmosphere, as in the regional Terrestrial Systems Modelling Platform (TSMP) (Shrestha et al., 2014; Gasper et al., 2014).*

*TSMP is a scale-consistent, highly modular, fully integrated soil-vegetation-atmosphere coupled regional climate model. TSMP comprises the hydrological model ParFlow v.3.2 (e.g., Kollet and Maxwell, 2008; Maxwell, 2013), the Community Land Model (CLM) v.3.5, and the atmospheric model Consortium for Small Scale Modelling (COSMO) v.5.01 (e.g., Baldauf et al., 2011), which are coupled externally via the Ocean Atmosphere Sea Ice Soil (OASIS, version 3.0) Model Coupling Toolkit (MCT) (e.g., Valcke, 2013) to exchange fluxes between independent component models of TSMP. Keune et al. (2016) demonstrated the link between groundwater and near-surface temperature in an analysis of the August 2003 European heat wave from the TSMP simulations nested within ERA-Interim (Dee et al., 2011) and set up over the the European domain of the COordinated Regional Downscaling EXperiment (EURO-CORDEX) (Gutowski et al., 2016; Jacob et al., 2020), with two different groundwater configurations: (i) simplified 1D free drainage approach and (ii) 3D physics-based variably saturated groundwater dynamics. The study clearly showed an impact of groundwater dynamics on the land surface water and energy balance: latent heat fluxes were higher and maximum temperatures were lower, especially in areas with shallow water table depth, in the 3D configuration compared to the simplified 1D free drainage approach. Keune et al. (2016) suggest that the 3D groundwater dynamics in TSMP alleviate the evolution of heat extremes due to weaker land-atmosphere feedbacks compared to the simplified 1D free drainage approach, at least during the investigated European heat wave of summer 2003. The ability of groundwater to decrease warm summer biases and moderate maximum air temperatures during a single seasonal heat wave in RCM simulations was also discussed in Barlage et al. (2015, 2021) and Mu et al. (2022).*

*As an explanation, the 3D groundwater dynamics in TSMP leads to shallower groundwater levels compared to 1D approach, causing wetter soils, and a reduction in the Bowen ratio (sensible heat flux to latent heat flux) due to an increase in surface latent heat flux and a decrease in surface sensible heat flux, i.e., an increase in evapotranspiration (Maxwell and Condon, 2016). On the one hand, such an increase in a latent heat flux causes moistening of the lower atmosphere and increases downward longwave radiation*

*due to the greenhouse effect of water vapor, on the other hand, it cools the surface and reduces outgoing surface longwave radiation (Pal and Eltahir, 2001). In addition, increased evapotranspiration may cause moist convection or rainfall, which further affects soil moisture (Eltahir, 1998; Yang et al., 2018). In its turn, the simplified representation of groundwater dynamics with the 1D free drainage approach leads to the opposite effect, namely an overestimation of the land surface-atmosphere coupling via shallow soil moisture and strengthening of the feedback mechanisms, i.e., deeper groundwater levels cause drier soils, an increase in the Bowen ratio by reducing latent and increasing sensible heat fluxes, a decrease in cloud cover and enhance of incoming shortwave radiation, and, as a result, higher near-surface temperatures, which in turn further enhances latent heat flux and reduces soil moisture (Vogel et al., 2018).*

*Further studies were carried out to understand whether the observed differences in simulated near-surface temperature due to differences in groundwater configuration (3D physics-based in TSMP and simplified in RCM ensemble) persist over longer time periods, and how this manifests itself for heat waves in the EURO-CORDEX realm. Furusho-Percot et al. (2019) showed that TSMP evaluation run (1996–2018) forced by the ERA-Interim reanalysis is able to capture climate system dynamics and the succession of warm and cold seasons on the regional scale for the PRUDENCE regions of Europe (Christensen and Christensen, 2007) consistently with E-OBS observations (Cornes et al., 2018). Another study by Furusho-Percot et al. (2022) demonstrated that TSMP multiannual simulations exhibit lower absolute deviations of summer heat wave indices from the E-OBS observational dataset, compared to ERA-Interim-driven RCM evaluation simulations of the EURO-CORDEX experiment (Jacob et al., 2020), which tend to simulate too persistent heat waves (Vautard et al., 2013). This particular behaviour of TSMP is attributed to the improved hydrology due to the explicit representation of 3D groundwater dynamics, namely the improved capacity to sustain soil moisture translates into more reliable latent heat flux and evapotranspiration, that, in turn, leads to a decrease in the heat wave amplitude, extent and the number of days with anomalously high near-surface temperatures, unlike in the CORDEX RCM ensemble with simplified groundwater representation. An important question remains: how will these findings be reflected in long-term regional climate simulations?.."*

5. The comparison between TSMP and other RCMs may not be entirely due to groundwater. Other factors such as forcing, and structure differences (how they model vegetation) may also contribute to the difference. An opportunity to address this is to run the TSMP without the groundwater component and compare the affected processes within TSMP rather than across different RCM settings.

We agree that the differences between the various RCMs and driving GCMs need to be honored in the analyses and discussion when making the comparison. At the beginning of Section 3.2, we discussed these limitations already:

*"Due to connections of various factors other than groundwater coupling in the multimodel CORDEX ensemble (e.g., various model setups, conceptual and structural model uncertainties, different physical parameterizations, internal variability, representation of*

*subsurface-land-atmosphere interactions, lower and lateral atmospheric GCM boundary conditions), it is challenging to reveal the exact cause and effect relationship of the explicit groundwater representation for simulated hot days and the associated heat events characteristics in RCMs. Moreover, the ensemble of EURO-CORDEX climate change scenario RCM control runs is not intended for direct comparison between individual models, as it includes different RCMs in combination with different driving GCMs. However, as has been shown in previous studies, the consideration of an extended period, e.g., 30-years, allows to draw statistically conclusions."*

To improve the manuscript further, we will (1) expand the discussion on the selection of RCMs-GCMs in the CORDEX ensemble, (2) clearly formulate the main objectives of the paper (please see our response to point #2 of "General remarks").

We would like to point out that Keune et al. (2016) with dedicated TSMP simulations (with and without 3D groundwater flow) clearly demonstrated the impact of groundwater on the land surface water and energy balance including temperature. In our study, the model was used in the same version (including groundwater) and with improved geology and topographic slopes. Repeating the dedicated simulations with/without groundwater at the climate time scale is computationally not feasible. Our rational and its limitations will be also discussed in detail in the revised manuscript.

**References**

[revised manuscript text omitted]

---

## Author Response (AR1)

**Reply to the editor**
**Groundwater in terrestrial systems modelling: a new climatology of extreme heat events in Europe**

*The authors are encouraged to submit a revised version, which will then be sent out for further evaluation.*

**We thank the editor for the encouragement to revise the manuscript. We have applied all changes according to our previously submitted responses to the reviewers. Please find the revised version of the manuscript together with the track-changes file.**

---

## Author Response (AR2)

**Reply to Referee #3**
**Impact of 3D groundwater dynamics on heat events in historical regional climate simulations over Europe**

**We thank the referee for the review and for the comments and suggestions. We provide a point-by-point reply below, where the reviewer comments are repeated in black. The replies to the reviewer's comments are in blue. The revised text is given in italics and in quotation marks.**

**General remarks**

- The study aims to improve understanding of how groundwater dynamics affect regional heat events by comparing the simulation results from TSMP and those from the CORDEX. In general, this is a well-organized paper. The topic and method are interesting.
  Thank you for the encouraging feedback.

- However, some issues shall be paid more attention. First, the research aim is not presented clearly in the abstract.
  Thank you for the suggestion. We revised the abstract of the paper and clearly stated the purpose of the study as suggested.
  *"By comparing summer heat events statistics (i.e. a series of consecutive days with a near-surface temperature exceeding the 90th percentile of the reference period) from TSMP and those from GCM-RCM simulations with simplified groundwater dynamics from the Coordinated Regional Climate Downscaling Experiment (CORDEX) for the European domain, we aim to improve the understanding of how 3D groundwater dynamics affect regional heat events over Europe."*

- It's a bit difficult to follow the results as the author mixed EURO-CORDEX, TSMP, and other GCM-RCM (e.g., REMO RCM, REMO2015 driven by IPSL-CM5A-LR, CCLM forced by MPI-ESM-LR).
  Thank you for your remark. We revised the text to ensure consistent notations, please see the revised version of the manuscript.

- In addition, since the study focuses on the impact of 3D groundwater dynamics on heat events, the related mechanisms are missing, which could better-assisting understanding the effect.
  Thank you for the comment. The introduction of the manuscript contains the paragraphs devoted to the description of the groundwater dynamics and related processes, as well as their relation to heat events based on previous studies (please see pp.3-4 in the manuscript). In addition, references to the TSMP modelling platform and its component models can be found throughout the text. Therefore, we believe that it gives the reader a sufficient overview of the subject of the paper.

**Specific Comments**

1. (Line 50, page 2) please give the full name of GCM and RCMs here when it is presented for the first from the main text.
   Thank you for the remark. We corrected the sentence as suggested.
   *"In the context of climate impact assessments, dynamical downscaling of global climate models (GCMs) with regional climate models (RCMs)..."*

2. (Line 130-145) References are missing in this part.
   The references were added as suggested.

3. (Line 255, page 10) "Hence the spatial pattern of the TG90p index significantly differs between different GCM-RCMs and the results from RCMs driven by the same GCMs show a rather similar behaviour." I found it challenging to get this result. Please explain it.
   Thank you for your comment. We rephrased this statement in the revised version of the manuscript.
   *"... the uncertainty in simulated near-surface temperature in summer is strongly controlled by the large-scale atmospheric circulation imposed by the GCM boundary conditions, with the largest impacts occurring in the southwestern PRUDENCE regions (e.g., Déqué et al., 2012; Evin et al., 2021). For this reason, the spatial pattern of the TG90p index in RCMs driven by the same GCMs show a rather similar behaviour."*

---

## Author Response (AR3)

**Reply to the editor**
**Impact of 3D groundwater dynamics on heat events in historical regional climate simulations over Europe**

**We thank the editor for the additional revision and the comments to improve the manuscript. We agree with the editor's suggestions and have revised the article accordingly.**

**Specific comments**

1. *There is too little discussion. The authors are recommended to include substantial in-depth discussion of their results, mechanisms, potential limitations, and implications.*

   Thank you for your recommendation. We have expanded the manuscript and added the discussion section in the revised version.

2. *I do not understand why on the summary section (line 433-438), the authors would list comparisons of numbers as bullet points. These should be presented more effectively in a table in the results section. The summary should highlight the most important take-home messages.*

   Thank you for the comment. The summary was rewritten in the revised version of the manuscript.